# Multi-Class Cost-Constrained Random Coding for Correlated Sources over the Multiple-Access Channel

**DOI:** 10.3390/e23050569

**Published:** 2021-05-03

**Authors:** Arezou Rezazadeh, Josep Font-Segura, Alfonso Martinez, Albert Guillén i Fàbregas

**Affiliations:** 1Department of Electrical Engineering, Chalmers University of Technology, SE-412 96 Gothenburg, Sweden; 2Department of Information and Communication Technologies, Universitat Pompeu Fabra, 08018 Barcelona, Spain; josep.font@upf.edu (J.F.-S.); alfonso.martinez@upf.edu (A.M.); albert.guillen@upf.edu (A.G.i.F.); 3Institució Catalana de Recerca i Estudis Avançats, 08010 Barcelona, Spain; 4Department of Engineering, University of Cambridge, Cambridge CB2 1PZ, UK

**Keywords:** multiple access channel, correlated sources, random coding, error exponents

## Abstract

This paper studies a generalized version of multi-class cost-constrained random-coding ensemble with multiple auxiliary costs for the transmission of *N* correlated sources over an *N*-user multiple-access channel. For each user, the set of messages is partitioned into classes and codebooks are generated according to a distribution depending on the class index of the source message and under the constraint that the codewords satisfy a set of cost functions. Proper choices of the cost functions recover different coding schemes including message-dependent and message-independent versions of independent and identically distributed, independent conditionally distributed, constant-composition and conditional constant composition ensembles. The transmissibility region of the scheme is related to the Cover-El Gamal-Salehi region. A related family of correlated-source Gallager source exponent functions is also studied. The achievable exponents are compared for correlated and independent sources, both numerically and analytically.

## 1. Introduction

In information theory, the fundamental problem of communication over a channel is studied from two complementary perspectives. First, one characterizes the transmissibility conditions, namely the circumstances under which the error probability asymptically vanishes as the blocklength goes to infinity. Second, one describes by means of error exponents the speed at which this error probability vanishes; the larger the exponent, the faster the error probability tends to zero. Since finding an exact expression for error probability is very difficult, a large body of work has investigated upper and lower bounds on the average error probability, or equivalently lower and upper bounds for the error exponent. In point-to-point, that is, single-user communication, using separate source-channel random coding [1,2], possibly with expurgation [1] (Eq. 5.7.10), yields lower bounds on the error exponent. In contrast, finding an upper bound to the error exponent satisfied by every code is more challenging. Generally, the hypothesis-testing method [3] is employed to derive upper bounds for the error exponent. Two well-known upper bounds to the error exponent are the sphere-packing exponent [4] and the minimum-distance exponent [5]. In fact, for rates greater than critical rate [1] (Sec. 5.6), the random-coding and sphere-packing bounds coincide with each other, while the expurgated and minimum-distance bounds coincide at rate zero.

For point-to-point communication, it was shown in ref. [1] (Prob. 5.16) that joint source-channel coding leads in general to a larger exponent than separate source-channel coding. Additionally, using codewords with a composition dependent on the source message leads to a larger exponent than the case where codewords are drawn according to a fixed product distribution [6,7]. Moreover, a scheme where source messages are assigned to disjoint classes and encoded by codes that depend on the class index, attains the sphere-packing exponent in those cases where it is tight [8].

Many works have been devoted to studying the transmissibility and the error exponent for a two-user multiple-access channel (MAC) [9,10,11]. Separate source-channel coding for the MAC with independent sources was studied in refs. [9,12]. In ref. [13], a universal exponent for the MAC was derived by considering separate source-channel coding. In ref. [14], a transmissible region is derived for the MAC under mismatched decoding, where the decoding rule is fixed and possibly suboptimal. In ref. [15], it was shown that using structure coding can improve the error exponent of the MAC. The maximum-error-probability criterion and the impact of feedback for the MAC were studied in ref. [16]. By considering separate source-channel coding, lower and upper bounds for the error exponent of the MAC were respectively obtained in refs. [17,18]. For the MAC with independent sources, the idea of exploting the dependency between messages and codewords was studied in ref. [19]. In ref. [20], an achievable exponent for the MAC with independent sources was given in the dual domain, that is, as a lower dimensional problem over parameters in terms of Gallager functions. For the MAC with correlated sources, it was shown in ref. [11] that considering statistical dependency between messages and codewords for the MAC with correlated sources leads to a larger transmissible region. However, an example presented in ref. [21] shows that one can reliably transmit information through the MAC without satisfying the reliable transmission obtained in ref. [11]. In another line of work, superposition coding with Gacs Körner Witsenhausen (GKW) common part is used in ref. [22] to to describe the sufficient conditions lossless recoverability.

In contrast to single-user communication, the problem of reliable transmission of two correlated sources has not been solved yet and just the sufficient conditions of a reliable transmission has been derived. In ref. [23], by applying coding techniques, a new set of sufficient conditions were proposed. Moreover, in ref. [24] new sufficient conditions for the three-user MAC with correlated sources were studied. In ref. [25], an achievable exponent derived was presented in the primal domain, that is, as a multi-dimensional optimization problem over distributions that is generally difficult to analyze.

In this paper, we examine how statistical dependency between the messages and codewords improves the exponent, as well as its impact on the transmissibility region. In view of refs. [1] (Ch. 7) and [26], we study a generalized message-dependent cost-constrained random-coding ensemble with multiple cost functions. By choosing the proper cost functions, the multi-class cost-constrained ensemble subsumes multiple ensembles previously considered in the literature and recovers the transmissibility region in ref. [11].

The paper is organized as follows—in Section 2, we present the problem of transmission of *N* correlated sources over an *N*-input discrete memoryless multiple-access channel and provide the key definitions of error probability, transmissibility, random-coding ensemble, and achievable exponent. In Section 3, we review the existing random-coding ensembles, define a novel generalized multi-class cost-constraint ensemble and characterize its achievable exponent. In the discussion Section 4, we characterize the transmissibility region for our error exponent, relate the exponent to standard Gallager source and channel functions, and provide numerical results and formulas that allow us to rank the exponents attained by the various standard random-coding ensembles.

## 2. Problem Formulation

We study the simultaneous transmission of *N* correlated, discrete, memoryless sources over a channel; users are indexed by ν∈N={1,2,…,N}. The source messages $u_{\nu }$ of user ν have *n* symbols drawn from the alphabet $U_{\nu }$. We denote by $u_{\sigma }$ the ordered vector of source messages for all users in a set $\sigma \subset 2^{N}$, i.e., a subset of the set of all user indices, and similarly by $U_{\sigma }$ the Cartesian product of the source alphabets in the set σ. When σ=N, $u_{N}$ and $U_{N}$ denote the ordered vector of source messages for all users and the Cartesian product of the all source alphabets respectively. The sources are memoryless and are characterized by the joint probability distribution $P_{N}$
(1)$$\begin{array}{c} P_{N}\left(u_{N}\right)=\prod_{t=1}^{n}P_{N}\left(u_{N,t}\right), \end{array}$$
and by the symbol joint probability distribution $P_{N}$. The source message and symbol marginal distributions of user ν∈N are denoted by $P_{\nu }$ and $P_{\nu }$ respectively. Assuming that the sources are independent, the marginal distributions induce new joint (mismatched) probability distributions of sets of users $\sigma \subset 2^{N}$. The induced independent-message and -symbol probabilities, denoted by $P_{\sigma }^{\mathrm{ind}}$ and $P_{\sigma }^{\mathrm{ind}}$, are given by
(2)$$\begin{array}{c} P_{\sigma }^{\mathrm{ind}}\left(u_{\sigma }\right)=\prod_{\nu \in \sigma }P_{\nu }\left(u_{\nu }\right), \end{array}$$
and similarly for $P_{\sigma }^{\mathrm{ind}}$.

Each user ν has an encoder that maps, without cooperation with the other users, the source message $u_{\nu }$ onto a codeword $x_{\nu }\left(u_{\nu }\right)$ also of length *n* and with symbols drawn from the alphabet $X_{\nu }$. We denote the codebook of user ν by $C_{n}^{\nu }$. We denote by $x_{\sigma }\in X_{\sigma }^{n}$ the vector of codewords for all users in a set $\sigma \subset 2^{N}$. Both terminals simultaneously send these codewords over a discrete memoryless multiple access channel with output alphabet Y. The symbolwise transition probability is denoted by *W*, and the channel is characterized by a conditional probability distribution
(3)$$\begin{array}{c} W\left(y|x_{N}\right)=\prod_{t=1}^{n}W\left(y_{t}|x_{N,t}\right), \end{array}$$
where y is the received sequence of length *n*.

Based on y, a joint decoder estimates all transmitted source messages $u_{N}$ according to the maximum a posteriori criterion:(4)$$\begin{array}{c} {\hat{u}}_{N}=\underset{u_{N}\in U_{N}^{n}}{arg max}P_{N}\left(u_{N}\right)W\left(y\;|\;x_{N}\left(u_{N}\right)\right), \end{array}$$
where $U_{N}^{n}$ denotes the set of all possible source messages $u_{N}$. An error occurs if the decoded messages ${\hat{u}}_{N}$ differ from the transmitted $u_{N}$; we refer to ${\hat{u}}_{N}\neq u_{N}$ as an error event. The error probability for a given set of codebooks, $P_{e}\left(C_{n}^{N}\right)$, is thus given by
(5)$$\begin{array}{c} P_{e}\left(C_{n}^{N}\right)≜\mathrm{Pr}\left\{{\hat{U}}_{N}\neq U_{N}\right\}. \end{array}$$
In our analysis, it will prove convenient to split the error event into $2^{N}-1$ distinct types of error events indexed by the non-empty subsets in the power set of the user indices $2^{N}\backslash \emptyset$, for example, τ∈{{1},{2},{1,2}} for N=2. More precisely, the error event of type τ corresponds to the conditions ${\hat{u}}_{\nu }\neq u_{\nu }$ for all ν∈τ and ${\hat{u}}_{\nu }=u_{\nu }$ for all $\nu \in \tau^{c}$, where $\tau^{c}$ is the complement of τ in the power set of the user indices.

We are interested in the asymptotics of the error probability for sufficiently large *n*, namely whether the error probability vanishes and how fast this probability tends to zero as it vanishes. The sources $U_{N}$ are said to be transmissible over the channel if there exists a sequence of codebooks $C_{n}^{N}$ such that $\lim_{n\to \infty }P_{e}\left(C_{n}^{N}\right)=0$. To characterize the speed at which the error probability vanishes, we use the notion of exponent. An exponent E is said to be achievable if there exists a sequence of codebooks such that
(6)$$\begin{array}{c} \underset{n\to \infty }{lim inf}-\frac{1}{n}\log P_{e}\left(C_{n}^{N}\right)\geq E. \end{array}$$

Source transmissibility and error-exponent achievability are typically studied by means of random coding. With random coding, one generates and studies sequences of ensembles of codebooks whose codewords are randomly drawn from a distribution $Q_{\nu }\left(x_{\nu }|u_{\nu }\right)$ independently for each user; as indicated by the notation, this distribution may possibly depend on the source message $u_{\nu }$. The random-coding probability distribution for the channel input $Q_{N}\left(x_{N}|u_{N}\right)$ combined for all users is given by
(7)$$\begin{array}{c} Q_{N}\left(x_{N}|u_{N}\right)=\prod_{\nu \in N}Q_{\nu }\left(x_{\nu }|u_{\nu }\right). \end{array}$$
The use of random coding allows us to study how the error probability averaged over the ensemble, denoted by ${\bar{P}}_{e}$ vanishes as *n* grows. More importantly, it shows the existence of good codes in the ensemble such that their error probability vanishes. For the point-to-point and the multiple-access channels, a number of such random-coding ensembles have been studied in the literature, as reviewed in the following section, where we also present a multi-class cost-constrained ensemble subsuming all these ensembles and characterize the achievable exponent and transmissibility region of this ensemble.

### Summary of Notation Used in the Paper

Sets are usually denoted by calligraphic upper case letter, e.g., X, and the *n*-Cartesian product set of X is denoted by $X^{n}$. The cardinality of a set such as X is denoted by |X|. The indicator function representing an error event or that an element *x* belongs to a set X is denoted by 1{x∈X}.

The number of users is denoted by *N* and user indices are typically represented by ν. The set of all users is denoted by N. The power set of all subsets of N is denoted by $2^{N}$ and the complement of a subset $\sigma \subset 2^{N}$ is denoted by $\sigma^{c}$; sets in the power set of users are denoted that by Greek letters, for example, τ and σ. The number of source-message classes and of cost functions for user ν are respectively denoted by $K_{\nu }$ and $L_{\nu }$; the sets of such classes are functions are respectively denoted by $K_{\nu }$ and $L_{\nu }$. Indices for source classes and cost functions are typically denoted by $i_{\nu }$ and $\ell_{\nu }$ respectively.

Subscripts and superscripts in a quantity *A* may represent sets of user indices σ. Depending on the context, the quantity represents a list or a suitable product of variables for all elements in the set σ. For instance, for σ={1,2}, $A^{\sigma }=\left(A^{1},A^{2}\right)$ or $A_{\sigma }=\left(A_{1},A_{2}\right)$. If the quantity is a probability distribution, its value for σ represents the probability distribution of the sequence, for example, $Q_{\sigma }^{i_{\sigma }}\left(x_{\sigma }\right)=\prod_{\nu \in \sigma }Q_{\nu }^{i_{\nu }}\left(x_{\nu }\right)$. If the quantity is a set, its value for σ is the Cartesian product, for example, $U_{\sigma }=U_{1}\times U_{2}$ for σ={1,2}. If σ=∅, then $A_{\sigma }=A^{\sigma }=0$. If σ is a singleton, for example, σ={2}, we simply write $A_{2}$ or $A^{2}$. We denote the operation that merges and sorts two lists $A_{\sigma_{1}}$ and $A_{\sigma_{2}}$ with $\sigma_{1}\cap \sigma_{2}=\emptyset$ into an ordered list containing all users in the union $\sigma_{1}\cup \sigma_{2}$ by $\left[A_{\sigma_{1}},A_{\sigma_{2}}\right]$. For sets of user indices, we denote such merging operation by $\left[\sigma_{1},\sigma_{2}\right]$ and we have $[\sigma ,\sigma^{c}]=N$.

Scalar random variables are denoted by capital letters, for example, *X* and lowercase letters represent a particular realisation, for example, x∈X. Capital bold letter denotes random vectors or sequences, for example, X, while small bold letter $x\in X^{n}$ denote deterministic vectors or sequences. Probability distributions for vectors or sequences, typically of length *n*, (resp. for symbols) are represented by text-style letters, for example, P, Q, W (resp. math-style letters, for example, *P*, *Q*, *W*). Sequences symbols are usually affixed a subscript to indicate a user index; the *t*-th symbol in the sequence $x_{\nu }$ is denoted by $x_{\nu ,t}$.

The source-symbol distribution for user ν is denoted by $P_{\nu }\left(u_{\nu }\right)$. The joint distribution for users σ is denoted by $P_{\sigma }\left(u_{\sigma }\right)$; the joint distribution, computed as if the sources were independent, is denoted by $P_{\sigma }^{\mathrm{ind}}\left(u_{\sigma }\right)$. The conditional source distribution for users $\sigma_{1}$ given another set $\sigma_{2}$ is denoted by $P_{\sigma_{1}|\sigma_{2}}\left(u_{\sigma_{1}}|u_{\sigma_{2}}\right)$. Vector or sequence distributions are defined analogously with *P* replaced by P. Channel input distributions are denoted by $Q_{\nu }\left(x_{\nu }\right)$, $Q_{\nu }^{i_{\nu }}\left(x_{\nu }\right)$, or $Q_{\nu ,u_{\nu }}^{i_{\nu }}\left(x_{\nu }\right)$, where $i_{\nu }$ denotes the index of the class source message and $Q_{\nu ,u_{\nu }}\left(x_{\nu }\right)$ is a shorthand for the conditional distribution $Q_{\nu }\left(x_{\nu }|u_{\nu }\right)$. Cost functions are similarly denoted by $a_{\nu }\left(x_{\nu }\right)$, $a_{\nu }^{i_{\nu }}\left(x_{\nu }\right)$, or $a_{\nu ,u_{\nu }}^{i_{\nu }}\left(x_{\nu }\right)$. Vector or sequence distributions are defined analogously with *Q* or *a* respectively replaced by Q or a. The conditional distribution for the channel output symbol (resp. sequence) is denoted by $W(y|x_{N})$ (resp. $W(y|x_{N})$).

## 3. Multi-Class Cost-Constrained Ensemble with Statistical Dependency

### 3.1. Review of Random-Coding Ensembles

The simplest and oldest random-coding ensemble is the independent, identically distributed (*iid*) [1,12,17,27], where the symbols $x_{\nu ,t}$ in all codewords $x_{\nu }$ of a given user ν are generated independently according to the same input distributions $Q_{\nu }\left(x_{\nu ,t}\right)$ for all source messages $u_{\nu }$. Throughout the paper, we shall identify ensembles by hyphenated acronyms, where the first part indicates the possible dependence of the codeword on the source message and the second part describes the generation of symbols in a codeword. This first ensemble is thus the message-independent iid (*mi-iid*) ensemble, since codewords have the same distribution for all source messages and symbols are independent of each other and independent of the source message symbols too. For the mi-iid ensemble, the random-coding distribution is given by
(8)$$\begin{array}{c} Q_{\nu }^{\mathrm{mi}-\mathrm{iid}}\left(x_{\nu }|u_{\nu }\right)=\prod_{t=1}^{n}Q_{\nu }\left(x_{\nu ,t}\right). \end{array}$$

In the message-independent, independent-conditionally-distributed (*mi-icd*) ensemble, the codewords $x_{\nu }$ of user ν are generated identically for all source messages $u_{\nu }$, independently of the full message $u_{\nu }$, and with symbols according to a set of $|\;U_{\nu }|$ conditional probability distributions $Q_{\nu ,u_{\nu }}\left(x_{\nu }\right)≜Q_{\nu }\left(x_{\nu }|u_{\nu }\right)$. To this end, let $I_{u_{\nu }}\left(u_{\nu }\right)$ denote the set of positions where the symbol $u_{\nu }\in U$ appears in the sequence $u_{\nu }$, namely
(9)$$\begin{array}{cc} I_{u_{\nu }}\left(u_{\nu }\right) & =\left\{t\in \left\{1,2,\ldots ,n\right\}:\;u_{\nu ,t}=u_{\nu }\right\}. \end{array}$$
Within each subsequence of $u_{\nu }$ where $u_{\nu ,t}=u_{\nu }$, represented by $u_{\nu }\left(I_{u_{\nu }}\left(u_{\nu }\right)\right)$, symbols are drawn independently according to $Q_{\nu ,u_{\nu }}\left(x_{\nu }\right)$. For this mi-icd ensemble, codewords are generated according to
(10)$$\begin{array}{cc} Q_{\nu }^{\mathrm{mi}-\mathrm{icd}}\left(x_{\nu }|u_{\nu }\right) & =\prod_{u_{\nu }\in U_{\nu }}\prod_{t\in I_{u_{\nu }}\left(u_{\nu }\right)}Q_{\nu ,u_{\nu }}\left(x_{\nu ,t}\right) \end{array}$$
(11)$$\begin{array}{cc} \; & =\prod_{t=1}^{n}Q_{\nu ,u_{\nu ,t}}\left(x_{\nu ,t}\right). \end{array}$$
Compared to the *mi-iid* ensemble, the *mi-icd* ensemble can lead to a larger transmissible region for the multiple-access channel with correlated sources [11,21]. An example of generation of three codewords $x_{\nu }^{\left(1\right)}$, $x_{\nu }^{\left(2\right)}$ and $x_{\nu }^{\left(3\right)}$ in the mi-icd ensemble is shown in Figure 1, for a given source sequence $u_{\nu }=\left(\alpha ,\beta ,\beta ,\gamma ,\beta ,\gamma ,\gamma ,\alpha ,\beta ,\alpha \right)$ with source alphabet U={α,β,γ}. To generate each codeword $x_{\nu }$ with alphabet X={a,c,e}, three subcodewords $x_{\nu }(I_{\alpha }\left(u_{\nu }\right)$, $x_{\nu }(I_{\beta }\left(u_{\nu }\right)$ and $x_{\nu }(I_{\gamma }\left(u_{\nu }\right)$ are pairwise-independently generated with i. i. d. distributions $Q_{\nu ,\alpha }=\left(1\backslash 3,1\backslash 3,1\backslash 3\right)$, $Q_{\nu ,\beta }=\left(1\backslash 2,1\backslash 4,1\backslash 4\right)$ and $Q_{\nu ,\gamma }=\left(1\backslash 3,2\backslash 3,0\right)$, respectively. Symbols generated according to $Q_{\nu ,\alpha }$, $Q_{\nu ,\beta }$ and $Q_{\nu ,\gamma }$ are respectively represented as green circles, blue boxes and red diamonds in the figure. In the example, $I_{\alpha }\left(u_{\nu }\right)=\left\{1,8,10\right\}$, $I_{\beta }\left(u_{\nu }\right)=\left\{2,3,5,9\right\}$ and $I_{\gamma }\left(u_{\nu }\right)=\left\{4,6,7\right\}$. For instance, the subcodeword $x_{\nu }^{\left(1\right)}(I_{\gamma }\left(u_{\nu }\right)$ has three symbols, each generated independently from $Q_{\nu ,\gamma }$, leading to the red-diamond symbols $x_{\nu }^{\left(1\right)}(I_{\gamma }\left(u_{\nu }\right)=\left(a,a,a\right)$.

Next, we have the message-dependent iid (*md-iid*) ensemble [6,8,19,25,28], where codewords for each user are generated with i. i. d. symbols according to different distributions $Q_{\nu }^{i_{\nu }}\left(x_{\nu }\right)$ that depend on the full source message through the class index $i_{\nu }$ of the class the source message belongs to. More precisely, for user ν=1,2 with source marginal distribution $P_{\nu }$, the $i_{\nu }$-th class $A_{\nu }^{i_{\nu }}$, where $i_{\nu }\in K_{\nu }=\left\{1,\ldots ,K_{\nu }\right\}$, is defined as the set of all source messages whose probability $P_{\nu }\left(u_{\nu }\right)$ is within a given interval, that is,
(12)$$\begin{array}{c} A_{\nu }^{i_{\nu }}=\left\{u_{\nu }\in U_{\nu }^{n}\;:\;\gamma_{\nu ,i_{\nu }}^{n}<P_{\nu }\left(u_{\nu }\right)\leq \gamma_{\nu ,i_{\nu }-1}^{n}\right\}, \end{array}$$
where the thresholds $\gamma_{\nu ,j}$ are $K_{\nu }+1$ non-negative numbers, ordered from higher to lower, such that $0=\gamma_{\nu ,K_{\nu }}\leq \gamma_{\nu ,K_{\nu }-1}\leq \ldots \leq \gamma_{\nu ,1}<\gamma_{\nu ,0}=1$, and $\min_{u_{\nu }}P_{\nu }\left(u_{\nu }\right)<\gamma_{\nu ,K_{\nu }-1}$ and $\gamma_{\nu ,1}\leq \max_{u_{\nu }}P_{\nu }\left(u_{\nu }\right)$. The *md-iid* random-coding distribution is given by
(13)$$\begin{array}{c} Q_{\nu }^{\mathrm{md}-\mathrm{iid}}\left(x_{\nu }|u_{\nu }\right)=\prod_{t=1}^{n}Q_{\nu }^{i_{\nu }\left(u_{\nu }\right)}\left(x_{\nu ,t}\right). \end{array}$$
The exponent of this *md-iid* ensemble can be larger than that of the *mi-iid* ensemble for joint source-channel coding [8,20,28].

In the message-dependent, independent conditional symbol distributions (*md-icd*) ensemble, messages in the class $i_{\nu }$ for user ν are encoded with codewords whose symbols are generated independently according to the conditional input distribution $Q_{\nu ,u_{\nu }}^{i_{\nu }}\left(x_{\nu }\right)$. The random-coding distribution of the *md-icd* ensemble is thus given by
(14)$$\begin{array}{c} Q_{\nu }^{\mathrm{md}-\mathrm{icd}}\left(x_{\nu }|u_{\nu }\right)=\prod_{u_{\nu }\in U_{\nu }}\prod_{t\in I_{u_{\nu }}\left(u_{\nu }\right)}Q_{\nu ,u_{\nu }}^{i_{\nu }\left(u_{\nu }\right)}\left(x_{\nu ,t}\right). \end{array}$$

In the message-independent, constant-composition (*mi-cc*) ensemble [29,30], codewords xν are drawn independently with an empirical distribution Q^ν(xν) close to a given Qν(xν), independently of the source message uν. For each user, codewords xν are randomly picked from Tνn(Qν), the set of all sequences whose empirical distribution has a variational distance to Qν of at most 1\n, that is
(15)Tνn(Qν)=xν∈Xνn:maxxνQ^ν(xν)−Qν(xν)<1n.
For this *mi-cc* ensemble, the random-coding distribution is given by
(16)$$\begin{array}{c} Q_{\nu }^{\mathrm{mi}-\mathrm{cc}}\left(x_{\nu }|u_{\nu }\right)=\frac{1}{|T_{\nu }^{n}\left(Q_{\nu }\right)|}1\left\{x_{\nu }\in T_{\nu }^{n}\left(Q_{\nu }\right)\right\}. \end{array}$$
While the *mi-cc* and *mi-iid* ensembles lead to identical transmissibility conditions, the former may achieve strictly larger exponents for suboptimal input distributions already in single-user settings [29].

The message-independent, conditional constant-composition (*mi-ccc*) ensemble combines features of the *mi-icd* and *mi-cc* ensembles. For each subsequence $u_{\nu }\left(I_{u_{\nu }}\left(u_{\nu }\right)\right)$, the corresponding subcodewords $x_{\nu }\left(I_{u_{\nu }}\left(u_{\nu }\right)\right)$ are drawn independently from the set $T_{\nu }^{|I_{u_{\nu }}\left(u_{\nu }\right)|}\left(Q_{\nu ,u_{\nu }}\right)$ of subsequences with empirical distribution close to $Q_{\nu ,u_{\nu }}\left(x_{\nu }\right)$, namely
(17)$$\begin{array}{c} T_{\nu }^{|I_{u_{\nu }}\left(u_{\nu }\right)|}\left(Q_{\nu ,u_{\nu }}\right)=\left\{x_{\nu }\in X_{\nu }^{|I_{u_{\nu }}\left(u_{\nu }\right)|}:\max_{x_{\nu }\in x_{\nu }}\;\left|{\hat{Q}}_{\nu }\left(x_{\nu }\right)-Q_{\nu ,u_{\nu }}\left(x_{\nu }\right)\right|<\frac{1}{|I_{u_{\nu }}\left(u_{\nu }\right)|}\right\}. \end{array}$$
The random-coding distribution of the mi-ccc ensemble is given by
(18)$$\begin{array}{c} Q_{\nu }^{\mathrm{mi}-\mathrm{ccc}}\left(x_{\nu }|u_{\nu }\right)=\prod_{u_{\nu }\in U_{\nu }}\frac{1}{\left|T_{\nu }^{|I_{u_{\nu }}\left(u_{\nu }\right)|}\left(Q_{\nu ,u_{\nu }}\right)\right|}1\left\{x_{\nu }\left(I_{u_{\nu }}\left(u_{\nu }\right)\right)\in T_{\nu }^{|I_{u_{\nu }}\left(u_{\nu }\right)|}\left(Q_{\nu ,u_{\nu }}\right)\right\}. \end{array}$$
An example of the generation of three codewords $x_{\nu }^{\left(4\right)}$, $x_{\nu }^{\left(5\right)}$ and $x_{\nu }^{\left(6\right)}$ in the mi-ccc ensemble is also shown in Figure 1 as a comparison to the md-iid ensemble, for the same source sequence $u_{\nu }$, source alphabet U={α,β,γ} and input alphabet X={a,c,e}. Now, to generate each codeword $x_{\nu }$, three subcodewords $x_{\nu }\left(I_{\alpha }\left(u_{\nu }\right)\right)$, $x_{\nu }\left(I_{\beta }\left(u_{\nu }\right)\right)$ and $x_{\nu }\left(I_{\gamma }\left(u_{\nu }\right)\right)$ are pairwise-independently, uniformly drawn in the type classes with empirical distributions ${\hat{Q}}_{\nu ,\alpha }$, ${\hat{Q}}_{\nu ,\beta }$ and ${\hat{Q}}_{\nu ,\gamma }$ that are closest to $Q_{\nu ,\alpha }$, $Q_{\nu ,\beta }$ and $Q_{\nu ,\gamma }$, respectively. Since in the example $|I_{\alpha }\left(u_{\nu }\right)|=3$, $|I_{\beta }\left(u_{\nu }\right)|=4$ and $|I_{\gamma }\left(u_{\nu }\right)|=3$, it follows that ${\hat{Q}}_{\nu ,\alpha }=\left(1\backslash 3,1\backslash 3,1\backslash 3\right)$, ${\hat{Q}}_{\nu ,\beta }=\left(1\backslash 2,1\backslash 4,1\backslash 4\right)$ and ${\hat{Q}}_{\nu ,\gamma }=\left(1\backslash 3,2\backslash 3,0\right)$. Symbols generated according to ${\hat{Q}}_{\nu ,\alpha }$, ${\hat{Q}}_{\nu ,\beta }$ and ${\hat{Q}}_{\nu ,\gamma }$ are respectively represented as green doubled circles, blue doubled boxes and red doubled diamonds in the figure. For instance, all subcodewords $x_{\nu }^{\left(j\right)}(I_{\gamma }\left(u_{\nu })\right)$, for j=4,5,6, have three symbols jointly generated from the constant-composition type ${\hat{Q}}_{\nu ,\gamma }$, that is, exactly one *a* and two *c*s.

The message-dependent, constant-composition (*md-cc*) ensemble combines the features of having different distributions for different messages with constant-composition random coding. For messages in the class $i_{\nu }\in \left\{1,\ldots ,K_{\nu }\right\}$ for user ν, codewords are drawn from the set of sequences with empirical distribution close to $Q_{\nu }^{i_{\nu }}\left(x_{\nu }\right)$. For this ensemble, the random-coding distribution is given by
(19)$$\begin{array}{c} Q_{\nu }^{\mathrm{md}-\mathrm{cc}}\left(x_{\nu }|u_{\nu }\right)=\frac{1}{\left|T_{\nu }^{n}\left(Q_{\nu }^{i_{\nu }\left(u_{\nu }\right)}\right)\right|}1\{x_{\nu }\in T_{\nu }^{n}(Q_{\nu }^{i_{\nu }\left(u_{\nu }\right)})\}. \end{array}$$

Finally, the message-dependent, conditional constant-composition (*md-ccc*) ensemble combines several of the ensembles listed above. For a given message $u_{\nu }=\left(u_{\nu ,1},\ldots ,u_{\nu ,n}\right)$ in the $i_{\nu }$-th class, that is, $u_{\nu }\in A_{\nu }^{i_{\nu }}$, the subsequence of $u_{\nu }$ having the same symbol $u_{\nu }$, that is, $u_{\nu }\left(I_{u_{\nu }}\left(u_{\nu }\right)\right)$, is encoded with pairwise-independent codewords generated from the set of codewords with empirical distribution very close to $Q_{\nu ,u_{\nu }}^{i_{\nu }}\left(x_{\nu }\right)$. The random-coding distribution of the md-ccc ensemble is thus given by
(20)$$\begin{array}{c} Q_{\nu }^{\mathrm{md}-\mathrm{ccc}}\left(x_{\nu }|u_{\nu }\right)=\prod_{u_{\nu }\in U_{\nu }}\frac{1}{\left|T_{\nu }^{|I_{u_{\nu }}\left(u_{\nu }\right)|}\left(Q_{\nu ,u_{\nu }}^{i_{\nu }\left(u_{\nu }\right)}\right)\right|}1\left\{x_{\nu }\left(I_{u_{\nu }}\left(u_{\nu }\right)\right)\in T_{\nu }^{|I_{u_{\nu }}\left(u_{\nu }\right)|}\left(Q_{\nu ,u_{\nu }}^{i_{\nu }\left(u_{\nu }\right)}\right)\right\}. \end{array}$$

### 3.2. Generalized Multi-Class Cost-Constrained Ensemble

Motivated by the ensembles listed in the previous section, and inspired by refs. [1] (Ch. 7) and [26] (Sec. II), we study a generalized message-dependent multi-class cost-constrained random-coding ensemble with multiple auxiliary costs.

For each user, we partition the set of source messages into $K_{\nu }$ disjoint classes with thresholds on the message probabilities as in Equation (Equation 12). Let the source message be in the $i_{\nu }$-th class, that is, $i_{\nu }\left(u_{\nu }\right)=i_{\nu }$. Given the source message $u_{\nu }$ and the source symbol $u_{\nu }$, we consider the subsequence $u_{\nu }\left(I_{u_{\nu }}\left(u_{\nu }\right)\right)$, where $I_{u_{\nu }}\left(u_{\nu }\right)$ is defined in Equation (Equation 9), and we denote the corresponding source subsequence and subcodeword by $u_{\nu }\left(I_{u_{\nu }}\left(u_{\nu }\right)\right)$ and $x_{\nu }\left(I_{u_{\nu }}\left(u_{\nu }\right)\right)$ respectively. For each user ν, class index $i_{\nu }$, and source message symbol $u_{\nu }$, the subcodeword $x_{\nu }\left(I_{u_{\nu }}\left(u_{\nu }\right)\right)$ is drawn according to a symbolwise i. i. d. distribution $Q_{\nu ,u_{\nu }}^{i_{\nu }}\left(x_{\nu }\right)$ conditioned on a set of cost constraints being satisfied. We consider $L_{\nu }$ additive cost functions $a_{\nu ,u_{\nu }}^{i_{\nu },\ell_{\nu }}\left(x_{\nu }\right)$, $\ell_{\nu }\in L_{\nu }=\left\{1,\ldots ,L_{\nu }\right\}$. The total cost $a_{\nu ,u_{\nu }}^{i_{\nu },\ell_{\nu }}\left(x_{\nu }\left(I_{u_{\nu }}\left(u_{\nu }\right)\right)\right)$ of the subcodeword $x_{\nu }\left(I_{u_{\nu }}\left(u_{\nu }\right)\right)$ is given by the sum of the symbol costs $a_{\nu ,u_{\nu }}^{i_{\nu },\ell_{\nu }}$, namely
(21)$$\begin{array}{c} a_{\nu ,u_{\nu }}^{i_{\nu },\ell_{\nu }}\left(x_{\nu }\left(I_{u_{\nu }}\left(u_{\nu }\right)\right)\right)=\sum_{j\in I_{u_{\nu }}\left(u_{\nu }\right)}a_{\nu ,u_{\nu }}^{i_{\nu },\ell_{\nu }}\left(x_{\nu ,j}\right). \end{array}$$
We assume that the average cost $\phi_{\nu ,u_{\nu }}^{i_{\nu },\ell_{\nu }}$ under the conditional distribution $Q_{\nu ,u_{\nu }}^{i_{\nu }}$ is zero:(22)$$\begin{array}{c} \phi_{\nu ,u_{\nu }}^{i_{\nu },\ell_{\nu }}=\sum_{x_{\nu }\in X_{\nu }}Q_{\nu ,u_{\nu }}^{i_{\nu }}\left(x_{\nu }\right)a_{\nu ,u_{\nu }}^{i_{\nu },\ell_{\nu }}\left(x_{\nu }\right)=0. \end{array}$$
Finally, fix some parameters $\delta_{\nu }>0$ and let $D_{\nu }^{i_{\nu }}$ be the set of codewords for which the average empirical cost of its constituent subcodewords $\frac{1}{|I_{u_{\nu }}\left(u_{\nu }\right)|}a_{\nu ,u_{\nu }}^{i_{\nu },\ell_{\nu }}\left(x_{\nu }\left(I_{u_{\nu }}\left(u_{\nu }\right)\right)\right)$ is close to the statistical mean $\phi_{\nu ,u_{\nu }}^{i_{\nu },\ell_{\nu }}=0$ for all cost functions and source symbols, i.e.,
(23)$$\begin{array}{c} D_{\nu ,u_{\nu }}^{i_{\nu }}≜\left\{x_{\nu }\in X_{\nu }^{n}:\left|\frac{1}{|I_{u_{\nu }}\left(u_{\nu }\right)|}a_{\nu ,u_{\nu }}^{i_{\nu },\ell_{\nu }}\left(x_{\nu }\left(I_{u_{\nu }}\left(u_{\nu }\right)\right)\right)\right|\leq \frac{\delta_{\nu }}{|I_{u_{\nu }}\left(u_{\nu }\right)|},\;u_{\nu }\in U_{\nu },\;\ell_{\nu }\in L_{\nu }\right\}. \end{array}$$

Codewords $x_{\nu }$ are the combination of subcodewords $x_{\nu }\left(I_{u_{\nu }}\left(u_{\nu }\right)\right)$ with respective positions in $I_{u_{\nu }}\left(u_{\nu }\right)$. For this multi-class cost-constrained ensemble, the random-coding distribution is thus given by
(24)$$\begin{array}{cc} Q_{\nu }^{\mathrm{cost}}\left(x_{\nu }|u_{\nu }\right)\; & =\frac{1}{\Xi_{\nu }}\prod_{u_{\nu }\in U_{\nu }}\prod_{t\in I_{u_{\nu }}\left(u_{\nu }\right)}Q_{\nu ,u_{\nu }}^{i_{\nu }}\left(x_{\nu ,t}\right)1\left\{x_{\nu }\in D_{\nu ,u_{\nu }}^{i_{\nu }}\right\} \end{array}$$
(25)$$\begin{array}{cc} \; & =\frac{1}{\Xi_{\nu }}\prod_{t=1}^{n}Q_{\nu ,u_{\nu ,t}}^{i_{\nu }}\left(x_{\nu ,t}\right)1\left\{x_{\nu }\in D_{\nu ,u_{\nu }}^{i_{\nu }}\right\}, \end{array}$$
where $\Xi_{\nu }$ is a normalizing constant and the class index is determined by the source message, $i_{\nu }=i_{\nu }\left(u_{\nu }\right)$.

The multi-class cost-constrained ensemble subsumes all the ensembles described in Section 3.1. First of all, the *iid* and *icd* ensembles are recovered by setting $L_{\nu }=0$ and choosing the appropriate number of classes $K_{\nu }$ and random-coding distributions $Q_{\nu }$, $Q_{\nu ,u_{\nu }}$, $Q_{\nu }^{i_{\nu }}$ and $Q_{\nu ,u_{\nu }}^{i_{\nu }}$. For all these cases, the set $D_{\nu ,u_{\nu }}^{i_{\nu }}$ includes all generated codewords and the normalizing constant is $\Xi_{\nu }=1$.

To recover the constant-composition ensembles, for which constraints force the subcodewords to belong to some set $T_{\nu }^{n}\left(Q_{\nu }\right)$ or $T_{\nu }^{|I_{u_{\nu }}\left(u_{\nu }\right)|}\left(Q_{\nu ,u_{\nu }}^{i_{\nu }}\right)$, for each of the $K_{\nu }$ classes for user ν we set $\delta_{\nu }<1$, $L_{\nu }=\left|X_{\nu }\right|$ and bijectively map the channel input symbols to cost function indices $\ell_{\nu }\left(x_{\nu }\right)$ so that
(26)$$\begin{array}{c} a_{\nu ,u_{\nu }}^{i_{\nu },\ell_{\nu }}\left(x_{\nu }\right)=1\left\{x_{\nu }=\ell_{\nu }\right\}-Q_{\nu ,u_{\nu }}^{i_{\nu }}\left(\ell_{\nu }\right). \end{array}$$
In case the ensemble does not depend on either $i_{\nu }$ or $u_{\nu }$, these symbols are dropped from Equation (Equation 26). For example, for the md-cc ensemble, we have $a_{\nu ,\ell_{\nu }}^{i_{\nu }}\left(\ell_{\nu }\right)=1\left\{x_{\nu }=\ell_{\nu }\right\}-Q_{\nu }^{i_{\nu }}\left(x_{\nu }\right)$. In addition, the codeword set $D_{\nu ,u_{\nu }}^{i_{\nu }}$ in Equation (Equation 23) is simplified as
(27)$$\begin{array}{c} D_{\nu ,u_{\nu }}^{i_{\nu }}=\left\{x_{\nu }\in X_{\nu }^{n}:\left|\frac{1}{n}\sum_{t=1}^{n}1\left\{x_{\nu ,t}=x\right\}-Q_{\nu }^{i_{\nu }}\left(x\right)\right|\leq \frac{1}{n},\;x\in X_{\nu }\right\}, \end{array}$$
which is the same as $T^{n}\left(Q_{\nu }^{i_{\nu }}\right)$ given a version of Equation (Equation 15) where $Q_{\nu }$ may depend on $i_{\nu }$.

Again, choosing the right number of classes $K_{\nu }$ and random-coding distributions $Q_{\nu }$, $Q_{\nu ,u_{\nu }}$, $Q_{\nu }^{i_{\nu }}$, and $Q_{\nu ,u_{\nu }}^{i_{\nu }}$ recovers the various constant-composition ensembles. By construction, the set $D_{\nu ,u_{\nu }}^{i_{\nu }}$ includes only the (sub)codewords with empirical distribution close to respectively $Q_{\nu }$, $Q_{\nu ,u_{\nu }}$, $Q_{\nu }^{i_{\nu }}$, and $Q_{\nu ,u_{\nu }}^{i_{\nu }}$, and the normalizing constant $\Xi_{\nu }$ is the probability of the corresponding type set (or product thereof). As an example, for the md-ccc ensemble, choosing the cost functions in Equation (Equation 26) as follows
(28)$$\begin{array}{c} a_{\nu ,u_{\nu }}^{i_{\nu },\ell_{\nu }}\left(x_{\nu }\left(I_{u_{\nu }}\left(u_{\nu }\right)\right)\right)=\sum_{j\in I_{u_{\nu }}\left(u_{\nu }\right)}1\left\{x_{\nu ,j}=\ell_{\nu }\right\}-Q_{\nu ,u_{\nu }}^{i_{\nu }}\left(\ell_{\nu }\right) \end{array}$$
yields the following cost-constraint set, which is equivalent to Equation (Equation 17),
(29)$$\begin{array}{c} D_{\nu ,u_{\nu }}^{i_{\nu }}=\left\{x_{\nu }\in X_{\nu }^{n}:|\frac{\sum_{j\in I_{u}\left(u_{\nu }\right)}1\left\{x_{\nu ,j}=x\right\}}{\left|I_{u}\left(u_{\nu }\right)\right|}-Q_{\nu ,u_{\nu }}^{i_{\nu }}\left(x\right)|\leq \frac{1}{\left|I_{u}\left(u_{\nu }\right)\right|},\;u\in U_{\nu },\;x\in X_{\nu }\right\}. \end{array}$$

### 3.3. Exponent for the Generalized Multi-Class Cost-Constrained Ensemble

**Theorem** **1.**
*For the transmission of N correlated memorlyess sources with joint distribution $P_{N}$, where N={1,2,…,N}, over a channel with input $x_{N}$ over a memoryless channel with transition probabilitiy $W(y|x_{N})$, consider a random-coding multi-class cost-constrained ensemble where source messages for each user ν∈N are allocated, depending on their probabilities, into $K_{\nu }$ classes with thresholds $\left\{\gamma_{\nu ,0},\gamma_{\nu ,1},\ldots ,\gamma_{\nu ,K_{\nu }}\right\}$, as in Equation (Equation 12), and encoded onto codewords randomly generated with a distribution $Q_{\nu }^{i_{\nu }}\left(x_{\nu }|u_{\nu }\right)$ that depends on the source message according to Equation (Equation 24) through symbol distributions $Q_{\nu ,u_{\nu }}^{i_{\nu }}$ that possibly depend on the source-message class index $i_{\nu }$ and source symbol $u_{\nu }$ and $L_{\nu }$ cost functions $a_{\nu ,u_{\nu }}^{i_{\nu },\ell_{\nu }}$, $\ell_{\nu }\in \left\{1,2,\ldots ,L_{\nu }\right\}$. This random-coding ensemble attains the following exponent $E^{\cos t}$*
(30)$$\begin{array}{cc} E^{\mathrm{cost}} & =\min_{\begin{array}{c} \tau \in 2^{N}\backslash \emptyset ,\;i_{N}\in K_{N} \end{array}}\;\max_{0\leq \rho \leq 1}\;\max_{\begin{array}{c} \lambda_{N}^{L,U}\geq 0,\;r_{Nu_{N}}^{\ell_{N}}\in R \end{array}}E_{\tau }^{i_{N}}\left(\rho ,\lambda_{N}^{L,U},r_{Nu_{N}}^{\ell_{N}}\right), \end{array}$$
*where the Gallager function $E_{\tau }^{i_{N}}\left(\rho ,\lambda_{N}^{L,U},r_{Nu_{N}}^{\ell_{N}}\right)$ is given by*
(31)$$\begin{array}{cc} \; & \;E_{\tau }^{i_{N}}\left(\rho ,\lambda_{N}^{L,U},r_{Nu_{N}}^{\ell_{N}}\right)=\\ \; & \;-\log\sum_{u_{\tau^{c}},x_{\tau^{c}},y}{\left(\sum_{u_{\tau },x_{\tau }}P_{N}{\left(u_{N}\right)}^{\frac{1}{1+\rho }}\Lambda_{N}^{i_{N}}\left(u_{N}\right)Q_{\tau ,u_{\tau }}^{i_{\tau }}\left(x_{\tau }\right)R_{N,u_{N}}^{i_{N}}\left(x_{N}\right){\left(Q_{\tau^{c},u_{\tau^{c}}}^{i_{\tau^{c}}}\left(x_{\tau^{c}}\right)W\left(y|x_{N}\right)\right)}^{\frac{1}{1+\rho }}\right)}^{\;\;1+\rho }, \end{array}$$
*and the functions $\Lambda_{\sigma }^{i_{\sigma }}\left(u_{\sigma }\right)$ and $R_{\sigma ,u_{\sigma }}^{i_{\sigma }}\left(x_{\sigma }\right)$ are respectively given by*
(32)$$\begin{array}{c} \Lambda_{\sigma }^{i_{\sigma }}\left(u_{\sigma }\right)=\prod_{\nu \in \sigma }{\left(\frac{P_{\nu }\left(u_{\nu }\right)}{\gamma_{\nu ,i_{\nu }}}\right)}^{\lambda_{\nu }^{L}}{\left(\frac{\gamma_{\nu ,i_{\nu }-1}}{P_{\nu }\left(u_{\nu }\right)}\right)}^{\lambda_{\nu }^{U}}, \end{array}$$
(33)$$\begin{array}{c} R_{\sigma ,u_{\sigma }}^{i_{\sigma }}\left(x_{\sigma }\right)=\prod_{\nu \in \sigma }\prod_{\ell_{\nu }\in L_{\nu }}e^{r_{\nu u_{\nu }}^{\ell_{\nu }}\;a_{\nu ,u_{\nu }}^{i_{\nu },\ell_{\nu }}\left(x_{\nu }\right)}, \end{array}$$
*and implicitly depende on the set of optimization parameters $\left(\lambda_{N}^{L,U},r_{Nu_{N}}^{\ell_{N}}\right)$.*


**Proof.** This result is proved in Appendix A. □

The random-coding exponent in Equation (Equation 30) depends on the partitioning of the source-message set into classes, the channel input distributions, and the codeword cost-constraint functions. The best possible generalized cost-constraint exponent is obtained by optimizing over the multi-class partitioning, the cost constraints and the input distributions. We briefly discuss the optimization w. r. t. the thresholds of the source messages partitioning in Appendix B. In the next section, we provide some numerical examples where we compute the optimal exponents for either independent or correlated sources, and find that the optimal number of classes is two. In ref. [31] (Sec. 3.2.1.1), we provide some indications of why this optimality of only two classes is harder to establish in multi-user scenarios, compared to the single-user case. In the next section, we use Equations (Equation 31) and (Equation 30) to respectively obtain the source and channel Gallager functions of the various ensembles in Section 3.1 and rank their achievable exponents and transmissibility regions.

## 4. Discussion

### 4.1. Gallager Functions for Correlated Sources

In this section, we evaluate the generalized Gallager function $E_{\tau }^{i_{N}}\left(\rho ,\lambda_{N}^{L,U},r_{Nu_{N}}^{\ell_{N}}\right)$ of the multi-class cost-constrained ensemble in Equation (Equation 31) for the various ensembles described in Section 3.1. In the cases where it is possible, we relate this Gallager function to the well-known [1] correlated-source and channel Gallager functions, respectively given by: (34)$$\begin{array}{c} E_{s,\sigma }\left(\rho ,P_{N}\right)=\log\sum_{u_{\sigma^{c}}}{\left(\sum_{u_{\sigma }}P_{N}{\left(u_{N}\right)}^{\frac{1}{1+\rho }}\right)}^{1+\rho }, \end{array}$$(35)$$\begin{array}{c} E_{0}\left(\rho ,Q,W\right)=-\log\sum_{y}{\left(\sum_{x}Q\left(x\right)W{\left(y|x\right)}^{\frac{1}{1+\rho }}\right)}^{1+\rho }, \end{array}$$
where $\sigma \in 2^{N}$. Using that $\left[u_{\sigma },u_{\sigma^{c}}\right]=u_{N}$, the standard Gallager source function is given by $E_{s}\left(\rho ,P_{N}\right)=E_{s,N}\left(\rho ,P_{N}\right)$, with N={1,…,N} the set of user indices.

For the simple mi-iid ensemble, with only one source class and no cost constraints, $K_{\nu }=1$ and $L_{\nu }=0$ for all ν∈N, and $\Lambda_{\sigma }^{i_{\sigma }}\left(u_{\sigma }\right)=R_{\sigma ,u_{\sigma }}^{i_{\sigma }}\left(x_{\sigma }\right)=1$ for all $\sigma \in 2^{N}$. With no statistical dependency between messages and codewords, $Q_{\nu ,u_{\nu }}\left(x_{\nu }\right)=Q_{\nu }\left(x_{\nu }\right)$. Setting $i_{N}=1$ and $\lambda_{N}^{L,U}=r_{Nu_{N}}^{\ell_{N}}=0$ in Equation (Equation 31) gives the Gallager function $E_{\tau }^{\mathrm{mi}-\mathrm{iid}}\left(\rho ,P_{N},Q_{N},W\right)$,
(36)$$\begin{array}{cc} E_{\tau }^{\mathrm{mi}-\mathrm{iid}} & \left(\rho ,P_{N},Q_{N},W\right)\\ \; & =-\log\sum_{u_{\tau^{c}},x_{\tau^{c}},y}{\left(\sum_{u_{\tau },x_{\tau }}P_{N}{\left(u_{N}\right)}^{\frac{1}{1+\rho }}Q_{\tau }\left(x_{\tau }\right){\left(Q_{\tau^{c}}\left(x_{\tau^{c}}\right)W\left(y|x_{N}\right)\right)}^{\frac{1}{1+\rho }}\right)}^{1+\rho }. \end{array}$$
Isolating the summations over $u_{\tau^{c}}$ and $u_{\tau }$, we can split the Gallager function as
(37)$$\begin{array}{cc} E_{\tau }^{\mathrm{mi}-\mathrm{iid}}\left(\rho ,P_{N},Q_{N},W\right) & =E_{0}\left(\rho ,Q_{\tau },Q_{\tau^{c}}W\right)-E_{s,\tau }\left(\rho ,P_{N}\right), \end{array}$$
where $Q_{\tau^{c}}W$ is a shorthand for $Q_{\tau^{c}}\left(x_{\tau^{c}}\right)W\left(y|x_{N}\right)$, the transition probability of a channel with input $x_{\tau }$ and output $\left(x_{\tau^{c}},y\right)$.

For the *mi-icd* ensemble, we have a similar set-up as for the *mi-iid* ensemble, where $Q_{\nu ,u_{\nu }}\left(x_{\nu }\right)$ now may depend on $u_{\nu }$. In this case, the Gallager function $E_{\tau }^{\mathrm{mi}-\mathrm{icd}}\left(\cdot \right)$ is given by Equation (Equation 36) with $Q_{\sigma }\left(x_{\sigma }\right)$ replaced by $Q_{\sigma ,u_{\sigma }}\left(x_{\sigma }\right)$, for $\sigma \in \{\tau ,\tau^{c}\}$:(38)$$\begin{array}{cc} E_{\tau }^{\mathrm{mi}-\mathrm{icd}} & \left(\rho ,P_{N},Q_{N,U},W\right)\\ \; & =-\log\sum_{u_{\tau^{c}},x_{\tau^{c}},y}{\left(\sum_{u_{\tau },x_{\tau }}P_{N}{\left(u_{N}\right)}^{\frac{1}{1+\rho }}Q_{\tau ,u_{\tau }}\left(x_{\tau }\right){\left(Q_{\tau^{c},u_{\tau^{c}}}\left(x_{\tau^{c}}\right)W\left(y|x_{N}\right)\right)}^{\frac{1}{1+\rho }}\right)}^{1+\rho }. \end{array}$$
As the summations over $u_{\tau^{c}}$ and $u_{\tau }$ are not independent from the rest, the Gallager function does not split into source and channel functions unless the sources are independent, in which case one can find an *mi-iid* ensemble with a tilted unconditional input distribution and identical exponent. To this end, and for a given conditional input distribution $Q_{\nu ,u_{\nu }}\left(x_{\nu }\right)$, let us define a tilted distribution $Q_{\nu }^{\rho }\left(x_{\nu }\right)$ as
(39)$$\begin{array}{c} Q_{\nu }^{\rho }\left(x_{\nu }\right)=\sum_{u_{\nu }}\frac{P_{\nu }{\left(u_{\nu }\right)}^{\frac{1}{1+\rho }}}{\sum_{{\bar{u}}_{\nu }}P_{\nu }{\left({\bar{u}}_{\nu }\right)}^{\frac{1}{1+\rho }}}Q_{\nu ,u_{\nu }}\left(x_{\nu }\right). \end{array}$$
From this equation, we have the following equality:(40)$$\begin{array}{c} Q_{\nu }^{\rho }\left(x_{\nu }\right)\sum_{{\bar{u}}_{\nu }}P_{\nu }{\left({\bar{u}}_{\nu }\right)}^{\frac{1}{1+\rho }}=\sum_{u_{\nu }}P_{\nu }{\left(u_{\nu }\right)}^{\frac{1}{1+\rho }}Q_{\nu ,u_{\nu }}\left(x_{\nu }\right). \end{array}$$
Substituting this identity together with $P_{N}\left(u_{N}\right)=P_{\tau }\left(u_{\tau }\right)P_{\tau^{c}}\left(u_{\tau^{c}}\right)$ in Equation (Equation 38) and rearranging the result, we obtain the following Gallager function for independent sources: (41)$$\begin{array}{cc} E_{\tau }^{\mathrm{mi}-\mathrm{icd}}\left(\rho ,P_{N},Q_{N,U},W\right) & =E_{0}\left(\rho ,Q_{\tau }^{\rho },WQ_{\tau^{c}}\right)-E_{s}\left(\rho ,P_{\tau }\right) \end{array}$$(42)$$\begin{array}{cc} \; & =E_{\tau }^{\mathrm{mi}-\mathrm{iid}}\left(\rho ,P_{N},\left[Q_{\tau }^{\rho },Q_{\tau^{c}}\right],W\right). \end{array}$$

For the *md-iid* and *md-icd* ensembles, there are $K_{\nu }$ source classes per user and no cost constraints, i.e., $L_{\nu }=0$ and $R_{\sigma ,u_{\sigma }}^{i_{\sigma }}\left(x_{\sigma }\right)=1$ for ν∈N and $\sigma \in 2^{N}$. Settting $r_{Nu_{N}}^{\ell_{N}}=0$ in Equation (Equation 31) gives the Gallager function $E_{\tau ,i_{N}}^{\mathrm{md}-\mathrm{icd}}\left(\cdot \right)$ for generic $i_{N}$ [31] (Eq. (4.36)),
(43)$$\begin{array}{cc} E & {}_{\tau ,i_{N}}^{\mathrm{md}-\mathrm{icd}}\left(\rho ,P_{N},Q_{N,U}^{i_{N}},W\right)\\ \; & =-\log\sum_{u_{\tau^{c}},x_{\tau^{c}},y}{\left(\sum_{u_{\tau },x_{\tau }}P_{N}{\left(u_{N}\right)}^{\frac{1}{1+\rho }}\Lambda_{N}^{i_{N}}\left(u_{{}_{N}}\right)Q_{\tau ,u_{\tau }}^{i_{\tau }}\left(x_{\tau }\right){\left(Q_{\tau^{c},u_{\tau^{c}}}^{i_{\tau c}}\left(x_{\tau^{c}}\right)W\left(y|x_{N}\right)\right)}^{\frac{1}{1+\rho }}\right)}^{1+\rho }. \end{array}$$

The Gallager function $E_{\tau ,i_{N}}^{\mathrm{md}-\mathrm{iid}}\left(\cdot \right)$ for the *md-iid* ensemble is obtained by setting $Q_{\sigma ,u_{\sigma }}\left(x_{\sigma }\right)=Q_{\sigma }\left(x_{\sigma }\right)$, independent of $u_{\nu }$, for $\sigma \in \{\tau ,\tau^{c}\}$ in Equation (Equation 43). As the summations over $u_{\tau^{c}}$ and $u_{\tau }$ are now independent from the rest, the Gallager function splits as
(44)$$\begin{array}{cc} \; & E_{\tau ,i_{N}}^{\mathrm{md}-\mathrm{iid}}\left(\rho ,P_{N},Q_{N}^{i_{N}},W\right)=E_{0}\left(\rho ,Q_{\tau }^{i_{\tau }},Q_{\tau^{c}}^{i_{\tau^{c}}}W\right)-E_{s,\tau }^{i_{N}}\left(\rho ,P_{N}\right), \end{array}$$
where we defined $E_{s,\tau }^{i_{N}}\left(\rho ,P_{N}\right)$, a modified Gallager $E_{s}$-function, as
(45)$$\begin{array}{c} E_{s,\tau }^{i_{N}}\left(\rho ,P_{N}\right)=\log\sum_{u_{\tau^{c}}}{\left(\sum_{u_{\tau }}P_{N}{\left(u_{N}\right)}^{\frac{1}{1+\rho }}\Lambda_{N}^{i_{N}}\left(u_{N}\right)\right)}^{1+\rho }. \end{array}$$
The maximization w.r.t. $\lambda_{N}^{L,U}$ in Equation (Equation 30) only affects the second term in the r. h. s. of Equation (Equation 44), since the function $\Lambda_{N}^{i_{N}}$ only appears in the source part of the exponent. In Appendix C, we discuss the properties of Equation (Equation 45) after the maximization w.r.t. $\lambda_{N}^{L,U}$ as a function of ρ, and establish some connections to the Gallager source function (Equation 34) and to the source functions for the single-user md-iid ensemble in ref. [8].

The Gallager functions for the constant-composition ensembles differ from the ones considered so far in the presence of $L_{\nu }=\left|X_{\nu }\right|$ cost functions $a_{\nu ,u_{\nu }}^{i_{\nu },\ell_{\nu }}\left(x_{\nu }\right)$, given in Equation (Equation 26), for each input distribution $Q_{\nu ,u_{\nu }}^{i_{\nu }}\left(x_{\nu }\right)$. These cost functions appear in the Gallager functions through the factors $R_{\sigma ,u_{\sigma }}^{i_{\sigma }}\left(x_{\sigma }\right)$, for $\sigma \in \{\tau ,\tau^{c}\}$ that multiply each appearance of $Q_{\sigma ,u_{\sigma }}^{i_{\sigma }}\left(x_{\sigma }\right)$ in the function, and through their associated optimization parameters $r_{Nu_{N}}^{\ell_{N}}$. The expressions of the Gallager functions for these constant-composition ensembles can be easily inferred from this obversation, so we focus on the factor $R_{\sigma ,u_{\sigma }}^{i_{\sigma }}\left(x_{\sigma }\right)$ itself.

For the *mi-cc* and *md-cc* ensembles, the cost functions $a_{\nu ,u_{\nu }}^{i_{\nu },\ell_{\nu }}\left(x_{\nu }\right)$, factor $R_{\sigma ,u_{\sigma }}^{i_{\sigma }}\left(x_{\sigma }\right)$, and associated optimization parameter $r_{\nu u_{\nu }}^{\ell_{\nu }}$ are independent of $u_{\nu }$, we thus write $a_{\nu }^{i_{\nu },\ell_{\nu }}\left(x_{\nu }\right)$, $R_{\sigma }^{i_{\sigma }}\left(x_{\sigma }\right)$, and $r_{\nu }^{\ell_{\nu }}$. The expressions in Equations (Equation 26) and (33) for $L_{\nu }=X_{\nu }$ give
(46)$$\begin{array}{c} R_{\nu }^{i_{\nu }}\left(x_{\nu }\right)=e^{\sum_{\ell_{\nu }\in X_{\nu }}r_{\nu }^{\ell_{\nu }}\left(1\left\{x_{\nu }=\ell_{\nu }\right\}-Q_{\nu }^{i_{\nu }}\left(\ell_{\nu }\right)\right)}. \end{array}$$
The exponent in Equation (Equation 46) can be evaluated as
(47)$$\begin{array}{cc} \sum_{\ell_{\nu }\in X_{\nu }}r_{\nu }^{\ell_{\nu }}\left(1\left\{x_{\nu }=\ell_{\nu }\right\}-Q_{\nu }^{i_{\nu }}\left(\ell_{\nu }\right)\right) & =r_{\nu }^{x_{\nu }}-\sum_{\ell_{\nu }\in X_{\nu }}r_{\nu }^{\ell_{\nu }}Q_{\nu }^{i_{\nu }}\left(\ell_{\nu }\right) \end{array}$$
(48)$$\begin{array}{cc} \; & =\alpha_{\tau ,\nu }^{i_{\nu }}\left(x_{\nu }\right), \end{array}$$
where we have defined a function $\alpha_{\tau ,\nu }^{i_{\nu }}\left(x_{\nu }\right)$ that depends on τ and $i_{\nu }$ through the optimization parameters $r_{\nu }^{\ell_{\nu }}$. We can be easily verify that $\alpha_{\tau ,\nu }^{i_{\nu }}$ has zero mean, in other words, $\sum_{x_{\nu }}\alpha_{\tau ,\nu }^{i_{\nu }}\left(x_{\nu }\right)Q_{\nu }^{i_{\nu }}\left(x_{\nu }\right)=0$. At this point, the parameters $r_{\nu }^{\ell_{\nu }}$ may be replaced by the equivalent real-valued functions $\alpha_{\tau ,\nu }^{i_{\nu }}\left(x_{\nu }\right)$. We obtain the *mi-cc* Gallager function $E_{\tau }^{\mathrm{mi}-\mathrm{cc}}\left(\cdot \right)$ by setting $i_{N}=1$ and $\lambda_{N}^{L,U}=0$ in Equation (Equation 31),
$$\begin{array}{l} E{}_{\tau }^{\mathrm{mi}-\mathrm{cc}}\left(\rho ,\alpha_{\tau ,N},P_{N},Q_{N},W\right) \end{array}$$
(49)$$\begin{array}{cc} \; & =-\log\sum_{u_{\tau^{c}},x_{\tau^{c}},y}{\left(\sum_{u_{\tau },x_{\tau }}P_{N}{\left(u_{N}\right)}^{\frac{1}{1+\rho }}Q_{\tau }\left(x_{\tau }\right)e^{\alpha_{\tau ,N}\left(x_{N}\right)}{\left(Q_{\tau^{c}}\left(x_{\tau^{c}}\right)W\left(y|x_{N}\right)\right)}^{\frac{1}{1+\rho }}\right)}^{1+\rho } \end{array}$$
(50)$$\begin{array}{cc} \; & =-\log\sum_{x_{\tau^{c}},y}{\left(\sum_{x_{\tau }}Q_{\tau }\left(x_{\tau }\right)e^{\alpha_{\tau ,N}\left(x_{N}\right)}{\left(Q_{\tau^{c}}\left(x_{\tau^{c}}\right)W\left(y|x_{N}\right)\right)}^{\frac{1}{1+\rho }}\right)}^{1+\rho }-E_{s,\tau }\left(\rho ,P_{N}\right), \end{array}$$
where we split the Gallager function into channel and source terms in analogy to Equation (Equation 37).

In ref. [31] (Eq. (4.49)), the *md-cc* ensemble was studied for N=2 users in both the primal and dual domains. The md-cc Gallager function $E_{\tau }^{\mathrm{md}-\mathrm{cc}}\left(\cdot \right)$ for *N* users is obtained by combining the derivation of Equation (Equation 50) with that of Equation (Equation 44) to yield
(51)$$\begin{array}{cc} E_{\tau }^{\mathrm{md}-\mathrm{cc}}\left(\rho ,\alpha_{\tau ,N}^{i_{N}},P_{N},Q_{N}^{i_{N}},W\right)=-\log & \sum_{x_{\tau^{c}},y}{\left(\sum_{x_{\tau }}Q_{\tau }\left(x_{\tau }\right)e^{\alpha_{\tau ,N}^{i_{N}}\left(x_{N}\right)}{\left(Q_{\tau^{c}}\left(x_{\tau^{c}}\right)W\left(y|x_{N}\right)\right)}^{\frac{1}{1+\rho }}\right)}^{1+\rho }\\ & -\log\sum_{u_{\tau^{c}}}{\left(\sum_{u_{\tau }}P_{N}{\left(u_{N}\right)}^{\frac{1}{1+\rho }}\Lambda_{N}^{i_{N}}\left(u_{{}_{N}}\right)\right)}^{1+\rho }. \end{array}$$
As in previous cases, the exponent is obtained after maximization over $\alpha_{\tau ,N}^{i_{N}}$.

Concluding our list, the cost functions $a_{\nu ,u_{\nu }}^{i_{\nu },\ell_{\nu }}\left(x_{\nu }\right)$, factors $R_{\sigma ,u_{\sigma }}^{i_{\sigma }}\left(x_{\sigma }\right)$, and parameters $r_{\nu u_{\nu }}^{\ell_{\nu }}$ for the *mi-ccc* and *md-ccc* ensembles do depend on $u_{\nu }$. In analogy to Equation (Equation 48), we define a zero-mean function $\beta_{\tau ,\nu ,u_{\nu }}^{i_{\nu }}\left(x_{\nu }\right)$ as
(52)$$\begin{array}{c} \beta_{\tau ,\nu ,u_{\nu }}^{i_{\nu }}\left(x_{\nu }\right)=r_{\nu u_{\nu }}^{x_{\nu }}-\sum_{\ell_{\nu }\in X_{\nu }}r_{\nu u_{\nu }}^{\ell_{\nu }}Q_{\nu ,u_{\nu }}^{i_{\nu }}\left(\ell_{\nu }\right), \end{array}$$
and similarly for $\beta_{\tau ,\nu ,u_{\nu }}\left(x_{\nu }\right)$ for the *mi-ccc* ensemble. The Gallager function for the *mi-ccc* ensemble $E_{\tau }^{\mathrm{mi}-\mathrm{ccc}}\left(\cdot \right)$ is obtained by combining the derivations of Equation (Equation 50) and of Equation (Equation 38),
(53)$$\begin{array}{c} E_{\tau }^{\mathrm{mi}-\mathrm{ccc}}\left(\rho ,\beta_{\tau ,N,u_{N}},P_{N},Q_{N,U},W\right)\\ =-\log\sum_{u_{\tau^{c}},x_{\tau^{c}},y}{\left(\sum_{u_{\tau },x_{\tau }}P_{N}{\left(u_{N}\right)}^{\frac{1}{1+\rho }}Q_{\tau ,u_{\tau }}\left(x_{\tau }\right)e^{\beta_{\tau ,N,u_{N}}\left(x_{N}\right)}{\left(Q_{\tau^{c},u_{\tau^{c}}}\left(x_{\tau^{c}}\right)W\left(y|x_{N}\right)\right)}^{\frac{1}{1+\rho }}\right)}^{1+\rho }. \end{array}$$
Similarly, for the *md-ccc* ensemble, and in agreeement with the 2-user case studied in ref. [31] (Eq. (4.45)), combining the derivations of Equations (Equation 50) and (Equation 43), yields
(54)$$\begin{array}{cc} \; & \;E_{\tau ,i_{N}}^{\mathrm{md}-\mathrm{ccc}}\left(\rho ,\beta_{\tau ,N,u_{N}}^{i_{N}},P_{N},Q_{N,U}^{i_{N}},W\right)\\ \; & \;=-\log\sum_{u_{\tau^{c}},x_{\tau^{c}},y}{\left(\sum_{u_{\tau },x_{\tau }}P_{N}{\left(u_{N}\right)}^{\frac{1}{1+\rho }}\Lambda_{N}^{i_{N}}\left(u_{{}_{N}}\right)Q_{\tau ,u_{\tau }}^{i_{\tau }}\left(x_{\tau }\right)e^{\beta_{\tau ,N,u_{N}}^{i_{N}}\left(x_{N}\right)}{\left(Q_{\tau^{c},u_{\tau^{c}}}^{i_{\tau c}}\left(x_{\tau^{c}}\right)W\left(y|x_{N}\right)\right)}^{\frac{1}{1+\rho }}\right)}^{1+\rho }. \end{array}$$

### 4.2. Transmissibility

We may obtain the transmissibility conditions from the achievable exponents derived in Section 4.1, following the random-coding method described in ref. [1] (Th. 5.6.4). The analysis extends the transmissibility condition for joint source-channel coding in ref. [1] (Prob. 5.16), to account for statistical dependency of the codeword on the source message in the multiuser set-up. As mentioned above, the source $U_{N}$ is transmissible over the channel *W* if there exists a sequence of codes with vanishing error probability, or equivalently, with strictly positive achievable error exponent $E^{cost}$ in Equation (Equation 30). As an example, we present the derivation for the mi-icd ensemble where the class and cost functions in Equations (Equation 32) and (33) are inactive, namely $\Lambda_{\sigma }^{i_{\sigma }}\left(u_{\sigma }\right)=R_{\sigma ,u_{\sigma }}^{i_{\sigma }}\left(x_{\sigma }\right)=1$ for all $\sigma \in 2^{N}$, and leave the general case of $K_{\nu }>1$ classes and cost-constrained codewords as an open problem.

For the *mi-icd* case, and similarly to Gallager’s $E_{0}$-function [1] (Th. 5.6.3), the Gallager function $E_{\tau }^{\mathrm{mi}-\mathrm{icd}}\left(\cdot \right)$ in Equation (Equation 38) is concave (∩) with respect to ρ and satisfies $E_{\tau }^{\mathrm{mi}-\mathrm{icd}}\left(\rho =0,\cdot \right)=0$. For every $\tau \subset 2^{N}\backslash \emptyset$, let ${\hat{\rho }}_{\tau }$ be the optimizer given by
(55)$${\hat{\rho }}_{\tau }=\underset{0\leq \rho \leq 1}{arg max}E_{\tau }^{\mathrm{mi}-\mathrm{icd}}\left(\rho ,P_{N},Q_{N,U},W\right).$$
Therefore, the achievable exponent is strictly positive, namely $E_{\tau }^{\mathrm{mi}-\mathrm{icd}}\left({\hat{\rho }}_{\tau },\cdot \right)>0$, as far as the slope of the $E_{\tau }^{\mathrm{mi}-\mathrm{icd}}\left(\rho ,\cdot \right)$ function is strictly positive at ρ=0, that is
(56)$$\frac{\partial }{\partial \rho }{\left.E_{\tau }^{\mathrm{mi}-\mathrm{icd}}\left(\rho ,P_{N},Q_{N,U},W\right)\right|}_{\rho =0}>0.$$
Taking the derivative with respect to ρ at both sides of Equation (Equation 38), after some algebraic manipulations, we find that (Equation 56) is equivalent to
(57)$$\begin{array}{cc} \sum_{u_{\tau^{c}}}P_{\tau^{c}}\left(u_{\tau^{c}}\right)\sum_{x_{\tau^{c}},y}\sum_{u_{\tau },x_{\tau }} & P_{\tau |\tau^{c}}\left(u_{\tau }|u_{\tau^{c}}\right)Q_{\tau ,u_{\tau }}\left(x_{\tau }\right)Q_{\tau^{c},u_{\tau^{c}}}\left(x_{\tau^{c}}\right)W\left(y|x_{N}\right)\times \\ \; & \times \log\frac{P_{\tau |\tau^{c}}\left(u_{\tau }|u_{\tau^{c}}\right)Q_{\tau^{c},u_{\tau^{c}}}\left(x_{\tau^{c}}\right)W\left(y|x_{N}\right)}{\sum_{{\bar{u}}_{\tau },{\bar{x}}_{\tau }}P_{\tau |\tau^{c}}\left({\bar{u}}_{\tau }|u_{\tau^{c}}\right)Q_{\tau ,{\bar{u}}_{\tau }}\left({\bar{x}}_{\tau }\right)Q_{\tau^{c},u_{\tau^{c}}}\left(x_{\tau^{c}}\right)W\left(y|\left[{\bar{x}}_{\tau },x_{\tau^{c}}\right]\right)}>0. \end{array}$$

We next write the expression in the left hand-side of the inequality (Equation 57) in terms of entropy and mutual information. We denote as H(P) the entropy of a source with distribution *P* [32] (Eq. (2.1)) and by I(Q,W) the mutual information of a channel *W* with input distribution *Q* [32] (Eq. (2.28)). For $\sigma \subset 2^{N}$, we define a channel input distribution $Q_{\tau |\sigma }$, that is conditioned to the source messages $u_{\sigma }$, as
(58)$$Q_{\tau |\sigma }\left(x_{\tau }|u_{\sigma }\right)=\sum_{u_{\tau }\in U_{\tau }}P_{\tau |\sigma }\left(u_{\tau }|u_{\sigma }\right)Q_{\tau ,u_{\tau }}\left(x_{\tau }\right).$$
Therefore, the transmissibility condition (Equation 57) can be compactly expressed as
(59)$$\begin{array}{c} H\left(P_{\tau |\tau^{c}}\right)<I\left(Q_{\tau |\tau^{c}},W|P_{\tau^{c}}Q_{\tau^{c}|\tau^{c}}\right),\;\tau \subset 2^{N}\backslash \emptyset . \end{array}$$
As it is, $Q_{\tau^{c}|\tau^{c}}$ is “transparent”, as it cancels inside the fraction, and the channel law may also be written as $Q_{\tau^{c}|\tau^{c}}W$, removing the conditioning in the mutual information. With N={1,2} in Equation (Equation 59), we recover the achievable Cover-El Gamal-Salehi region [11] (Eq. (3)).

### 4.3. Numerical Examples

In this section, we present two simple examples showing that the exponent of the md-iid ensemble can be larger than that of the mi-iid ensemble with only two classes (and associated input distributions) for each user. First, we consider two correlated discrete memoryless sources, N=2 and N={1,2}, with alphabet $U_{\nu }=\left\{0,1\right\}$ for both users ν∈N, and probability distribution $P_{N}\left(u_{1},u_{2}\right)$ given in matrix form as
(60)$$P_{N}=\left(\begin{array}{cc} 0.0005 & 0.0095\\ 0.0005 & 0.9895 \end{array}\right).$$
The sources are sent over a discrete memoryless multiple-access channel with input alphabets $X_{1}=X_{2}=\left\{1,2,3,4,5,6\right\}$ and output alphabet Y={1,2,3,4}. The channel transition probabilites are given by a 36 × 4 matrix *W*, such that $W(y|x_{1},x_{2})$ is the row $x_{1}+6\left(x_{2}-1\right)$. The transition matrix *W* is given by
(61)$$\begin{array}{c} W=\left(\begin{array}{c} W_{1}\\ W_{2}\\ W_{3}\\ W_{4}\\ W_{5}\\ W_{6} \end{array}\right), \end{array}$$
where the 6 × 4 submatrices $W_{\ell }$, ℓ=1,…,6 are given as follows. First, the submatrix $W_{1}$ corresponds to the point-to-point channel discussed in ref. [8] (Sec. IV.C), given by
(62)$$\begin{array}{c} W_{1}=\left(\begin{array}{cccc} 1-3k_{1} & k_{1} & k_{1} & k_{1}\\ k_{1} & 1-3k_{1} & k_{1} & k_{1}\\ k_{1} & k_{1} & 1-3k_{1} & k_{1}\\ k_{1} & k_{1} & k_{1} & 1-3k_{1}\\ 0.5-k_{2} & 0.5-k_{2} & k_{2} & k_{2}\\ k_{2} & k_{2} & 0.5-k_{2} & 0.5-k_{2} \end{array}\right), \end{array}$$
for $k_{1}=0.045$ and $k_{2}=0.01$. Let the *m*-th row of matrix $W_{1}$ is denoted by $W_{1}\left(m\right)$. The matrix $W_{2}$ (resp. $W_{3}$) is a 6×4 matrix whose rows are all $W_{1}\left(5\right)$ (resp. $W_{1}\left(6\right)$). The matrices $W_{4}$, $W_{5}$ and $W_{6}$ are respectively given by
(63)$$\begin{array}{c} W_{4}=\left(\begin{array}{c} W_{1}\left(2\right)\\ W_{1}\left(3\right)\\ W_{1}\left(4\right)\\ W_{1}\left(1\right)\\ W_{1}\left(6\right)\\ W_{1}\left(5\right) \end{array}\right),\;W_{5}=\left(\begin{array}{c} W_{1}\left(3\right)\\ W_{1}\left(4\right)\\ W_{1}\left(1\right)\\ W_{1}\left(2\right)\\ W_{1}\left(5\right)\\ W_{1}\left(6\right) \end{array}\right),\;W_{6}=\left(\begin{array}{c} W_{1}\left(4\right)\\ W_{1}\left(1\right)\\ W_{1}\left(2\right)\\ W_{1}\left(3\right)\\ W_{1}\left(6\right)\\ W_{1}\left(5\right) \end{array}\right). \end{array}$$

The optimal achievable exponent [8] (Sec. IV.C) for the single-user channel $W_{1}$ in Equation (Equation 62) is related to two different distributions $Q^{☆}$ and $Q^{†}$, given in vector form by
(64)$$\begin{array}{c} Q^{☆}=\left(0,0,0,0,1\backslash 2,1\backslash 2\right), \end{array}$$
(65)$$\begin{array}{c} Q^{†}=\left(1\backslash 4,1\backslash 4,1\backslash 4,1\backslash 4,0,0\right). \end{array}$$
We let each user employ these distributions in the md-iid ensemble with input distribution in Equation (Equation 13) according to the source message partitioning in Equation (Equation 12) with $K_{\nu }=2$ classes per user and thresholds $\gamma_{N}=\left(\gamma_{1},\gamma_{2}\right)$. Since we consider two input distributions for each user, the channel Gallager function $\max_{\rho \in [0,1]}E_{0}\left(\rho ,Q_{\tau }^{i_{\tau }},WQ_{\tau^{c}}^{i_{\tau^{c}}}\right)$ is not concave in ρ [8]. To find the *md-iid* exponent $E^{\mathrm{md}-\mathrm{iid}}$, we optimize over the class thresholds following the method in Appendix B with the Gallager function in Equation (Equation 44), exploit the properties of the source function in Equation (Equation 45) in Appendix C, and also find the optimal input distribution assignment of $Q_{\nu }^{i_{\nu }}$ for each ν∈{1,2}. In our setting, we have four possible assignments, namely
(66)$$\begin{array}{c} \Omega_{1}\;:\;Q_{1}^{1}=Q_{2}^{1}=Q^{☆},\;Q_{1}^{2}=Q_{2}^{2}=Q^{†}, \end{array}$$
(67)$$\begin{array}{c} \Omega_{2}\;:\;Q_{1}^{1}=Q_{2}^{2}=Q^{☆},\;Q_{1}^{2}=Q_{2}^{1}=Q^{†}, \end{array}$$
(68)$$\begin{array}{c} \Omega_{3}\;:\;Q_{1}^{2}=Q_{2}^{1}=Q^{☆},\;Q_{1}^{1}=Q_{2}^{2}=Q^{†}, \end{array}$$
(69)$$\begin{array}{c} \Omega_{4}\;:\;Q_{1}^{2}=Q_{2}^{2}=Q^{☆},\;Q_{1}^{1}=Q_{2}^{1}=Q^{†}. \end{array}$$

We start our numerical discussion by assessing which of the possible four assignments in Equations (Equation 66)–(69) leads to a higher error exponent. For each possible pair of thresholds $\left(\gamma_{1},\gamma_{2}\right)$, we numerically calculate the optimal assignment $\Omega^{☆}\left(\gamma_{N}\right)$ given by
(70)$$\Omega^{☆}\left(\gamma_{N}\right)=\underset{\Omega_{j}}{arg max}\min_{i_{N}}\min_{\tau }E_{\tau }^{i_{N}}\left(\gamma_{N}\right),$$
and the corresponding achievable error exponent $E^{md-iid}\left(\gamma_{N}\right)$ as
(71)$$E^{cost}\left(\gamma_{N}\right)=\max_{\Omega_{j}}\min_{i_{N}}\min_{\tau }E_{\tau }^{i_{N}}\left(\gamma_{N}\right),$$
where the exponent function $E_{\tau }^{i_{N}}\left(\gamma_{N}\right)$ is given in Equation (Equation 133). Figure 2 and Figure 3 respectively show $\Omega^{☆}\left(\gamma_{N}\right)$ and $E^{cost}\left(\gamma_{N}\right)$ for the valid range of $\gamma_{N}$. For most pair of thresholds $\left(\gamma_{1},\gamma_{2}\right)$, assignments $\Omega_{1}$ and $\Omega_{3}$ lead to the highest exponent among the possible assignments, while assignments $\Omega_{2}$ and $\Omega_{4}$ are optimal only for a marginal region. Using this information, and combined with the values of the achievable exponents in Figure 3, we determine the message-dependent exponent
(72)$$E^{md-iid}=\max_{\gamma_{N}}E^{cost}\left(\gamma_{N}\right).$$

In this example, we obtained the achievable exponent $E^{\mathrm{md}-\mathrm{iid}}=0.2611$, corresponding to the input distribution assignment $\Omega_{1}$ in Equation (Equation 66) and optimal source message partitioning $\gamma_{1}^{☆}=0.8469$ and $\gamma_{2}^{☆}=0.6581$. The optimal point $\gamma_{N}^{☆}$ is shown by a white (black) bullet in Figure 2 (Figure 3).

Alternatively, we may first optimize over $\gamma_{N}$ and then over the assignments $\Omega_{j}$. To do so, we solve the system of Equation (Equation 136) in Appendix B to numerically determine the optimal thresholds $\gamma_{N}^{☆}$, and compute the exponent $E^{\mathrm{cost}}\left(\Omega_{j}\right)$ as
(73)$$E^{\mathrm{cost}}\left(\Omega_{j}\right)=\min_{i_{N}}\min_{\tau }E_{\tau }^{i_{N}}\left(\gamma_{N}^{☆}\right),$$
where the exponent function $E_{\tau }^{i_{N}}\left(\gamma_{N}\right)$ is given in Equation (Equation 133). We provide in Table 1 the values of the optimal thresholds $\gamma_{N}^{☆}$ and exponents $E_{\tau }^{i_{N}}\left(\gamma_{N}^{☆}\right)$ under the different assignment $\Omega_{j}$, for the three types of error τ and the four possible user classes $i_{N}$. For each assignment, the minimum over $i_{N}$ and τ as in Equation (Equation 73) is highlighted in gray, leading to the exponent $E^{\cos t}\left(\Omega_{j}\right)$. The message-dependent exponent is then
(74)$$\begin{array}{c} E^{\mathrm{md}-\mathrm{iid}}=\max_{j}E^{\cos t}\left(\Omega_{j}\right), \end{array}$$
recovering the error exponent $E^{\mathrm{md}-\mathrm{iid}}=0.2611$ for input distribution assignment $\Omega_{1}$ obtained using the previous method in Equation (Equation 71).

In the second example, we consider the transmission of two independent discrete memoryless sources with identical source alphabets $U_{\nu }=\left\{0,1\right\}$ with distributions induced by the marginals of Equation (Equation 60), given by $P_{1}\left(0\right)=0.01$ and $P_{2}\left(0\right)=0.001$. These sources are transmitted over the multiple-access channel with transition probability given by Equation (Equation 61), and are encoded using the md-iid ensemble with the input distribution assignments $\Omega_{j}$ in Equations (Equation 66)–(69). Following the footsteps of the correlated sources case, in Table 2 we calculate optimal thresholds $\gamma_{N}^{☆}$ and exponents $E_{\tau }^{i_{N}}\left(\gamma_{N}^{☆}\right)$ for the possible input distribution assignments and determine the exponent of the md-iid ensemble using Equations (Equation 73) and (Equation 74). In this case, the optimal assignment is again $\Omega_{1}$, with optimal source message partitioning specified by the thresholds $\gamma_{1}^{☆}=0.8779$ and $\gamma_{2}^{☆}=0.6933$, achieving an exponent of $E^{\mathrm{md}-\mathrm{iid}}=0.2458$, slightly smaller than that of correlated sources.

For the sake of completeness and purpose of comparison, we also calculate the exponent for the *mi-iid* ensemble described in Equation (Equation 8). In the absence of message dependence, for a given assignment $\Omega_{j}$, the *mi-iid* exponent is given by
(75)$$\begin{array}{c} E^{\mathrm{no}-\mathrm{cost}}\left(\Omega_{j}\right)=\min_{\tau }E_{\tau }, \end{array}$$
where the exponent function $E_{\tau }$ is given by $E_{\tau }=\max_{\rho }E_{\tau }^{\mathrm{mi}-\mathrm{iid}}\left(\rho ,P_{N},Q_{N},W\right)$ and $E_{\tau }^{\mathrm{mi}-\mathrm{iid}}$ is the Gallager function in Equation (Equation 37), described in the previous subsection. For both the correlated and independent sources described above, Table 3 presents the achievable exponents $E_{\tau }$ for each type of error τ and input distribution assignment $\left(Q_{1},Q_{2}\right)$, where $Q_{1}$ and $Q_{2}$ are either of $Q^{☆}$ and $Q^{†}$ in Equations (Equation 64) and (65). In our numerical example for correlated sources, the assignment with highest exponent is $\left(Q_{1},Q_{2}\right)=\left(Q^{†},Q^{☆}\right)$, giving an exponent of $E^{\mathrm{mi}-\mathrm{iid}}=0.2503$, slightly smaller than that of the md-iid ensemble. In contrast, the mi-iid exponent for independent sources, according to the second part of Table 3 is found to be $E^{\mathrm{mi}-\mathrm{iid}}=0.2367$ with input distribution $\left(Q_{1},Q_{2}\right)=\left(Q^{☆},Q^{†}\right)$. In this case, the md-iid exponent $E^{\mathrm{md}-\mathrm{iid}}$ is around 4% larger that the *mi-iid*; this situation is in contrast with to-point communication, where the gain in exponent achieved by an ensemble with two distributions is typically smaller, for example, 1% in ref. [8]. Hence, message-dependent random coding with two class distributions, compared to iid random coding, may lead to a higher error exponent gain in the MAC than in point-to-point communication.

### 4.4. Comparison of the Random-Coding Achievable Error Exponents

From the numerical results presented in Section 4.3, as well as from refs. [8,20,28,31], the message-dependent ensembles attain in general a larger exponent than their message-independent counterparts. We now compare the random-coding exponents for the ensembles presented in Section 3.1, whose Gallager functions were obtained in Section 4.1.

For independent sources, we found in Equation (42) that for a given conditional input distribution $Q_{\nu ,u_{\nu }}\left(x_{\nu }\right)$ and ρ, there exists an *iiid* distribution $Q_{\nu ,\rho }$ given by Equation (Equation 39) with identical Gallager function. Thus, the *mi-iid* and *mi-icd* ensembles attains the same exponent, after maximization over the input distributions. Similarly, we conclude that *md-iid* and *md-icd*-ensembles attain the same exponent.

In ref. [31] (Prop. 2.9), it was proved that for point-to-point communication, the exponent of the mi-ccc ensemble may be lower than that of the mi-cc ensemble. The same steps actually prove the same result for the MAC with independent sources. Thus, for the MAC with independent sources we have
(76)$$\begin{array}{c} E^{mi-ccc}\leq E^{mi-cc}\leq E^{md-cc},\;E^{md-ccc}\leq E^{md-cc}, \end{array}$$
(77)$$\begin{array}{c} E^{mi-iid}\leq E^{mi-cc}\leq E^{md-cc},\;E^{md-iid}\leq E^{md-cc}, \end{array}$$
and $E^{md-cc}$ is thus largest among the ensembles in Section 3.1 for an arbitrary input distribution. As discussed in ref. [29] (Th. 4), for optimal input distributions both $E^{md-cc}$ and $E^{md-iid}$ may coincide.

Concerning the optimal partitioning into message classes, for point-to-point communication it is known that partitioning the source-message set into two classes is sufficient to attain the optimal error exponent [8,31] (Prop. 2.7). However, the proof of ref. [31] (Prop. 2.7) cannot be easily generalized to the MAC with independent sources. At the same time, we could not find an example showing that assigning more than two input distributions leads to a larger exponent. Hence, finding the sufficient number of input distributions is for the message-dependent exponent is an open problem.

The comparisons in Equations (Equation 76) and (77) for correlated sources require, in general, a more sophisticated machinery and we consider here two simple cases. For the message-dependent *md-icd* and *md-ccc* ensembles, we observe that compared to $E_{\tau ,i_{N}}^{md-icd}$ in Equation (Equation 43) the $E_{\tau ,i_{N}}^{md-ccc}$ exponent in Equation (Equation 54) contains an additional term $\beta_{\tau ,N,u_{N}}^{i_{N}}\left(x_{N}\right)$ to guarantee the constant-composition distribution as in Equation (Equation 52). This allows to recover $E_{\tau ,i_{N}}^{md-ccc}$ by setting $\beta_{\tau ,N,u_{N}}^{i_{N}}\left(x_{N}\right)=0$ in $E_{\tau ,i_{N}}^{md-icd}$ and to prove that $E^{md-icd}\leq E^{md-ccc}$ after maximizing w. r. t. $\beta_{\tau ,N,u_{N}}^{i_{N}}(x_{N})$. Similarly for the ensembles with statistical independence between messages and codewords, we observe that the constant-composition exponent $E_{\tau ,i_{N}}^{md-cc}$ in Equation (Equation 51) also contains the additional term $\alpha_{\tau ,N}^{i_{N}}\left(x_{N}\right)$ compared to its *iid* counterpart $E_{\tau ,i_{N}}^{md-iid}$ in Equation (Equation 44), yielding $E^{md-iid}\leq E^{md-cc}$. Put together, for correlated sources it holds that
(78)$$\begin{array}{c} E^{md-icd}\leq E^{md-ccc},\;E^{md-iid}\leq E^{md-cc}, \end{array}$$
suggesting that, as in the case of single-user communication, the use of constant-composition input distributions may lead to higher exponents than the symbol-wise independent distributions when transmitting correlated sources over the MAC.

Summarizing, proper choices of the cost functions recover the different coding schemes considered in Section 3.1, including message-dependent and message-independent versions of iid, independent conditionally distributed, constant-composition, and conditional constant composition ensembles. Thanks to the flexibility of the generalized cost-constraint random-coding ensemble, the achievable exponents of the various ensembles can be compared and ranked, both numerically and analytically. 

## Figures and Tables

**Figure 1 entropy-23-00569-f001:**
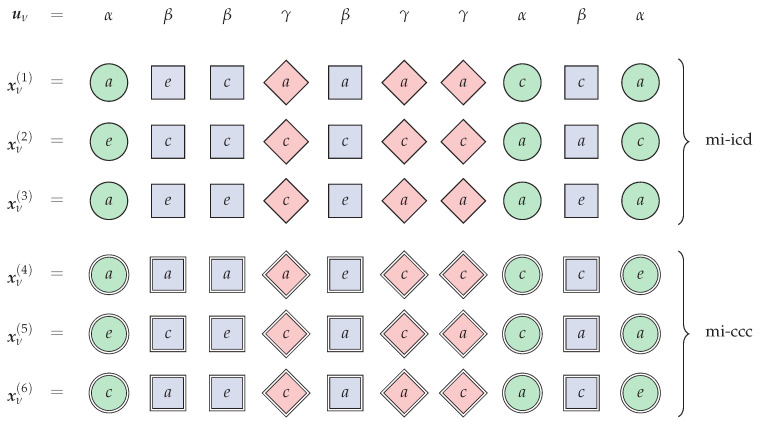
Example of codewords $x_{\nu }^{\left(1\right)}$, $x_{\nu }^{\left(2\right)}$ and $x_{\nu }^{\left(3\right)}$ in the mi-icd ensemble and $x_{\nu }^{\left(4\right)}$, $x_{\nu }^{\left(5\right)}$ and $x_{\nu }^{\left(6\right)}$ in the mi-ccc ensemble, for a given source sequence $u_{\nu }$.

**Figure 2 entropy-23-00569-f002:**
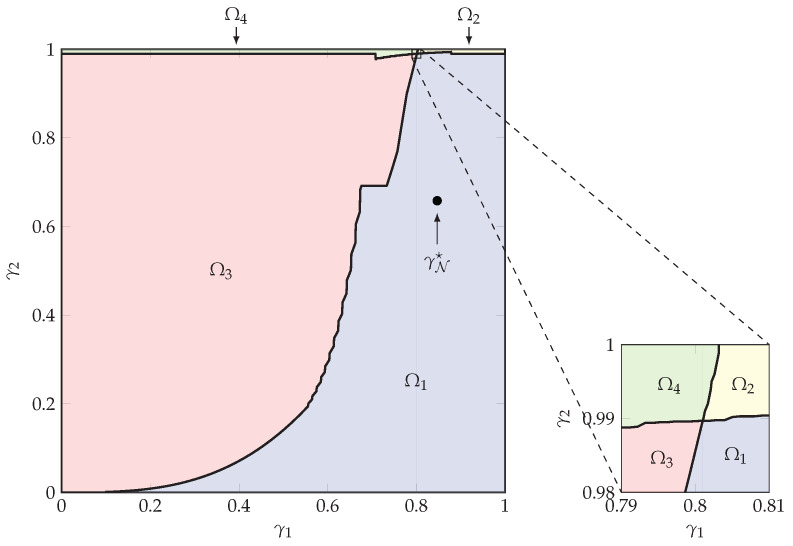
Correlated-sources optimal assignment $\Omega^{☆}\left(\gamma_{N}\right)$ in Equation (Equation 70) for all pairs of thresholds $\left(\gamma_{1},\gamma_{2}\right)$.

**Figure 3 entropy-23-00569-f003:**
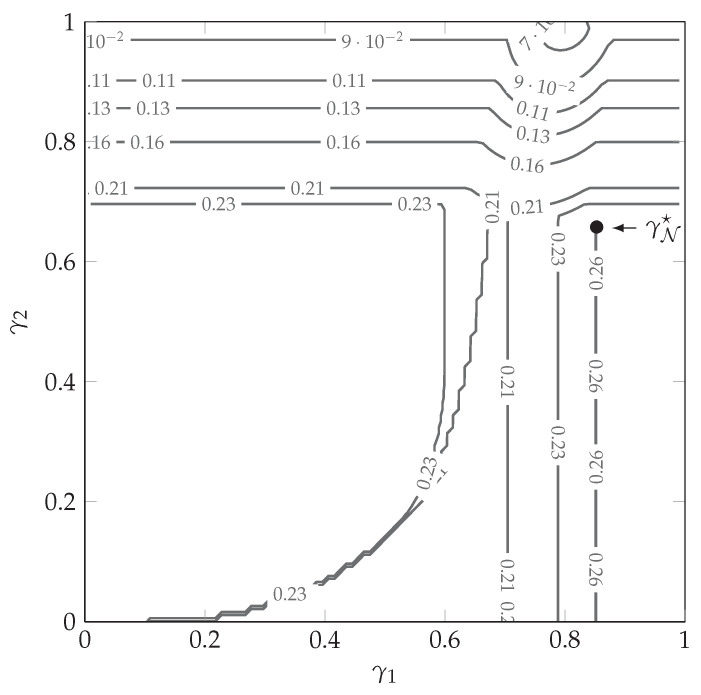
Correlated-sources error exponent $E^{cost}\left(\gamma_{N}\right)$ in Equation (Equation 71) for all pairs of thresholds $\left(\gamma_{1},\gamma_{2}\right)$.

**Table 1 entropy-23-00569-t001:** Correlated-sources optimal thresholds $\gamma_{N}^{☆}$ and exponents $E_{\tau }^{i_{N}}\left(\gamma_{N}^{☆}\right)$ in Equation (Equation 73) for assignments $\Omega_{j}$ in Equations (Equation 66)–(69). For each assignment, the minimum over $i_{N}$ and τ is highlighted in gray.

	Assignment $\Omega_{1}$					Assignment $\Omega_{2}$			
	$\gamma_{1}^{☆}=0.8469$ $\gamma_{2}^{☆}=0.6581$					$\gamma_{1}^{☆}=1$ $\gamma_{2}^{☆}=1$			
$\left(i_{1},i_{2}\right)$	(1,1)	(1,2)	(2,1)	(2,2)		(1,1)	(1,2)	(2,1)	(2,2)
τ={1}	0.3131	0.2735	0.3120	0.2611		0.0642	0.3268	0.1005	0.3604
τ={2}	0.3986	0.4369	0.2611	0.4119		0.3959	0.3986	0.4323	0.3110
τ={1,2}	0.2611	0.2972	0.2630	0.2883		0.2108	0.2108	0.2360	0.2637
	**Assignment** $\Omega_{3}$					**Assignment** $\Omega_{4}$			
	$\gamma_{1}^{☆}=0.5605$ $\gamma_{2}^{☆}=0.6709$					$\gamma_{1}^{☆}=0.6985$ $\gamma_{2}^{☆}=0.9033$			
$\left(i_{1},i_{2}\right)$	(1,1)	(1,2)	(2,1)	(2,2)		(1,1)	(1,2)	(2,1)	(2,2)
τ={1}	0.3120	0.2503	0.2763	0.2897		0.0879	0.3605	0.0879	0.3112
τ={2}	0.2503	0.3898	0.5675	0.5731		0.3664	0.2503	0.4720	0.4684
τ={1,2}	0.2630	0.2816	0.2503	0.3012		0.2360	0.2632	0.2097	0.2097

**Table 2 entropy-23-00569-t002:** Independent-sources md-iid optimal thresholds $\gamma_{N}^{☆}$ and exponents $E_{\tau }^{i_{N}}\left(\gamma_{N}^{☆}\right)$ in Equation (Equation 73) for assignments $\Omega_{j}$ in Equations (Equation 66)–(69). For each assignment, the minimum over $i_{N}$ and τ is highlighted in gray.

	Assignment $\Omega_{1}$					Assignment $\Omega_{2}$			
	$\gamma_{1}^{☆}=0.8779$ $\gamma_{2}^{☆}=0.6933$					$\gamma_{1}^{☆}=0.8776\;$ $\gamma_{2}^{☆}=1$			
$\left(i_{1},i_{2}\right)$	(1,1)	(1,2)	(2,1)	(2,2)		(1,1)	(1,2)	(2,1)	(2,2)
τ={1}	0.3343	0.2458	0.3089	0.2458		0.0913	0.3341	0.0913	0.3089
τ={2}	0.3850	0.3987	0.2458	0.3788		0.4555	0.3850	0.4357	0.2459
τ={1,2}	0.2730	0.2870	0.2685	0.2863		0.3430	0.2728	0.2956	0.2685
	**Assignment** $\Omega_{3}$					**Assignment** $\Omega_{4}$			
	$\gamma_{1}^{☆}=0.61$ $\gamma_{2}^{☆}=0.7043$					$\gamma_{1}^{☆}=0.7092$ $\gamma_{2}^{☆}=1$			
$\left(i_{1},i_{2}\right)$	(1,1)	(1,2)	(2,1)	(2,2)		(1,1)	(1,2)	(2,1)	(2,2)
τ={1}	0.3089	0.2367	0.2681	0.3078		0.0913	0.3117	0.0913	0.2648
τ={2}	0.2367	0.3672	0.5425	0.5538		0.4269	0.2367	0.5393	0.4683
τ={1,2}	0.2685	0.2811	0.2367	0.3133		0.3006	0.2685	0.2740	0.2164

**Table 3 entropy-23-00569-t003:** Mi-iid exponents $E_{\tau }$ in Equation (Equation 75) for two correlated and two independent sources vs several input distribution assigments $\left(Q_{1},Q_{2}\right)$. For each assignment, the minimum over τ is highlighted in gray.

	Correlated Sources			
$\left(Q_{1},Q_{2}\right)$	$\left(Q^{☆},Q^{☆}\right)$	$\left(Q^{☆},Q^{†}\right)$	$\left(Q^{†},Q^{☆}\right)$	$\left(Q^{†},Q^{†}\right)$
τ={1}	0.2682	0.0642	0.3120	0.0879
τ={2}	0.3986	0.3986	0.2503	0.3696
τ={1,2}	0.2097	0.2097	0.2630	0.2360
	**Independent Sources**			
$\left(Q_{1},Q_{2}\right)$	$\left(Q^{☆},Q^{☆}\right)$	$\left(Q^{☆},Q^{†}\right)$	$\left(Q^{†},Q^{☆}\right)$	$\left(Q^{†},Q^{†}\right)$
τ={1}	0.2648	0.3089	0.0627	0.0865
τ={2}	0.3850	0.2367	0.3850	0.3559
τ={1,2}	0.2164	0.2685	0.2164	0.2421

## Data Availability

Data is contained within the article.

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
