# Peer review of "Multi-Class Cost-Constrained Random Coding for Correlated Sources over the Multiple-Access Channel"

_entropy, 2021, doi:10.3390/e23050569_

Round 1
Reviewer 1 Report
The complex theoretical approach is well described and novel.
However:
(1) the need for the study is not strongly underlined in the introduction, nor the novelty
(2) beyond the authors papers, the references are at least 3 years old;
(3) a conclusion section underlining the main achievements, limitations and future work is missing.
Reviewer 2 Report
- I suggest the title of Figure 1 should be shortened. Reference should be made to Figure 1 in the text of the paper, and then it should be described what it shows, rather than doing so in the title of the figure.
- The title of Chapter 4 should be changed in accordance with the description of that part of the paper in the introduction.
- In “Table 2: Independent-sources md-iid optimal thresholds γ ? N and exponents E iN τ (γ ? N ) in (73) for assignments Ωj (66)–(69)”, on p. 17, the following should be corrected: “Table 2: Independent-sources md-iid optimal thresholds γ ? N and exponents E iN τ (γ ? N ) in (73) for assignments Ωj in (66)–(69)”.
- I believe that the paper lacks an exact conclusion and recommendations for further research.
The paper deals with an interesting issue. It is well written and organized. The results are presented in a very systematic way.
The paper can be published after entering the proposed corrections.
